# Global chromatin mobility induced by a DSB is dictated by chromosomal conformation and defines the HR outcome

**Fabiola García Fernández[†‡], Etienne Almayrac[†], Ànnia Carré Simon, Renaud Batrin, Yasmine Khalil, Michel Boissac, Emmanuelle Fabre***

Université de Paris, IRSL, INSERM, U944, CNRS, UMR7212, Paris, France

**Abstract** Repair of DNA double-strand breaks (DSBs) is crucial for genome integrity. A conserved response to DSBs is an increase in chromatin mobility that can be local, at the site of the DSB, or global, at undamaged regions of the genome. Here, we address the function of global chromatin mobility during homologous recombination (HR) of a single, targeted, controlled DSB. We set up a system that tracks HR in vivo over time and show that two types of DSB-induced global chromatin mobility are involved in HR, depending on the position of the DSB. Close to the centromere, a DSB induces global mobility that depends solely on H2A(X) phosphorylation and accelerates repair kinetics, but is not essential. In contrast, the global mobility induced by a DSB away from the centromere becomes essential for HR repair and is triggered by homology search through a mechanism that depends on H2A(X) phosphorylation, checkpoint progression, and Rad51. Our data demonstrate that global mobility is governed by chromosomal conformation and differentially coordinates repair by HR.

**\*For correspondence:**
emmanuelle-g.fabre@inserm.fr

[†]These authors contributed
equally to this work

**Present address:** [‡]Institut Curie,
CNRS, UMR3664, Sorbonne
Université F- 75005, Paris, France

**Competing interest:** The authors
declare that no competing
interests exist.

**Reviewing Editor:** Irene E
Chiolo, University of Southern
California, United States

## Editor's evaluation

This study is of relevance to the field of DNA repair. It uses a cleverly designed new recombination assay in yeast to address the impact of DNA break position on global genome mobility. A centromere–proximal DNA double–strand break (DSB) induces an H2A(X) phosphorylation–dependent global mobility that accelerates but is not essential for DSB repair, while a centromere–distal DSB triggers global mobility that is essential for repair and which depends on H2A(X) phosphorylation, Rad9 and Rad51. Together, these data support a model where global genome mobility promotes homologous recombination repair, particularly for centromere–distal DSBs, and help settle some recent controversy in the field.

## Introduction

Double-strand breaks (DSBs) are harmful lesions that constantly challenge the eukaryotic genome. DSBs can lead to deleterious genetic alterations such as modifications of the DNA at the DSB site (insertions/deletions) or global chromosomal rearrangements (translocations). To faithfully maintain genome stability and survive DSBs, cells display a DNA damage response (DDR) involving complex networks that detect, signal, and repair DSBs. Two of the most conserved factors contributing to DSB signaling are the checkpoint kinases Tel1 and Mec1 (mammalian ATM/ATR orthologues) that catalyze the phosphorylation of histone H2A in yeast and of H2AX in mammalian cells, forming γ-H2A(X) (*Jackson and Bartek, 2009*; *Waterman et al., 2020*). γ-H2A(X) extends widely on both sides of the

DSB, up to 100 kb in yeast and 1 MB in mammals (*Aymard et al., 2017*; *Caron et al., 2012*; *Iacovoni et al., 2010*; *Rogakou et al., 1999*; *Shroff et al., 2004*). This wide spreading of γ-H2A(X) serves as a docking site for further DDR proteins such as the mediator protein Rad9 (53BP1 in mammals) (*Kinner et al., 2008*).

DSBs can be accurately repaired by homologous recombination (HR). This mode of repair, predominant in S phase, uses the sister chromatid as template (*Symington and Gautier, 2011*). When homologous sequences are not found in the close vicinity of the break, a global genomic homology search may occur. Homology search starts with the formation of a presynaptic nucleoprotein filament that is composed of 3'-ssDNA overhangs coated with the recombinase protein Rad51 (*Chen et al., 2008*; *Kalocsay et al., 2009*; *Sugawara et al., 2003*; *Wang and Haber, 2004*). The Rad51 nucleofilament enables the sampling and recognition of homology within the nucleus through base pairing reviewed in *Kaniecki et al., 2018*. By analyzing Rad51 distribution in haploid yeast after the induction of a single DSB by chromatin immunoprecipitation (ChIP), a study visualized 'snapshots' of ongoing homology search, and found Rad51 strikingly distributed over a large portion of the broken chromosome (*Renkawitz et al., 2013*). Notably, the genome-wide Rad51 signal distribution was concomitant with that of Mec1-dependent γ-H2A(X), suggesting that Mec1 might be attached on the probing end during homology search. The resulting phosphorylation of H2A(X) could thus promote chromatin remodeling, signaling, or repair.

The non-random organization of chromosomes in the nucleus affects numerous nuclear processes including homology search. In the configuration of budding yeast chromosomes (known as the Rabl configuration), centromeres are clustered close to the yeast microtubule organizing center (spindle pole body), as evidenced by abundant *trans*-contacts between centromeres, and telomeres are bound to the nuclear periphery in a position that depends on the size of the chromosome arms (*Duan et al., 2010*; *Jin et al., 2000*; *Schober et al., 2008*; *Therizols et al., 2010*). It was shown in yeast that DSBs recombine more efficiently in spatially proximal regions (*Agmon et al., 2013*; *Batté et al., 2017*; *Lee et al., 2015*). Recombination between homologous sequences that lie close to centromeres is more efficient than between regions that are distantly located on chromosome arms (*Agmon et al., 2013*). Accordingly, induction of a DSB close to a centromere triggers concomitant Rad51 and γ-H2A(X) signals within centromeric regions of all other yeast chromosomes, suggesting that homology search can efficiently occur on DNA that is located proximal to the centromere of other chromosomes (*Lee et al., 2014*; *Renkawitz et al., 2013*). Likewise, mammalian topologically associated domains (TADs), where chromatin contacts are enriched, promote diffusion of ATM, which allows DSB signaling and thus repair (*Arnould et al., 2021*).

The search for distantly located homologous regions challenges recombination, likely requiring additional sophisticated mechanisms, such as chromatin mobility. Originally observed in yeast, increased chromatin mobility, both at the damaged locus (local mobility) and elsewhere in the undamaged genome (global mobility), is a conserved response to DSBs (reviewed in *García Fernández and Fabre, 2022*; *Oshidari et al., 2020*; *Seeber et al., 2018*; *Smith and Rothstein, 2017*). Different mechanisms have been described to explain these DSB-induced chromatin dynamics, ranging from intrinsic changes in chromatin properties, including chromatin stiffening or decompaction, to extrinsic nuclear organizing factors such as actin and microtubules (*Caridi et al., 2018*; *Hauer et al., 2017*; *Herbert et al., 2017*; *Lamm et al., 2020*; *Lawrimore et al., 2017*; *Miné-Hattab et al., 2017*; *Strecker et al., 2016*). Functionally, the mobility of damaged and undamaged chromatin has been implicated in various DDR processes such as DSB relocation, clustering, and homology search. Several studies have proposed that both types of mobility allow homology probing in a larger nuclear volume (*Dion et al., 2012*; *Miné-Hattab and Rothstein, 2012*; *Seeber et al., 2013*; *Smith et al., 2018*). During homology search, spreading of Rad51 and γ-H2A(X) take place concomitantly and the increase in local and global mobility is abolished in the absence of Rad51 or γ-H2A(X) (*García Fernández et al., 2021*; *Miné-Hattab et al., 2017*; *Miné-Hattab and Rothstein, 2012*; *Smith et al., 2018*). The role of global mobility in HR remains to be understood. Conflicting results show either a positive correlation between global mobility and recombinant products formation, or a lack of effect of global mobility on recombination in yeast (*Challa et al., 2021*; *Cheblal et al., 2020*; *Dion et al., 2012*; *Hauer et al., 2017*; *Miné-Hattab et al., 2017*; *Miné-Hattab and Rothstein, 2012*; *Strecker et al., 2016*). Similarly, mobility within a mammalian TAD, allowing ATM to reach other regions of chromatin, has also been suggested to explain the favorable environment for HR observed within a TAD (*Aymard et al., 2017*;

*Schrank et al., 2018*). Although the mechanisms and functions of increased mobility are not fully understood, it is now considered part of the DDR.

To characterize the role of global chromatin mobility upon DNA damage during HR in yeast, we set up a system in haploid cells that allows us to simultaneously monitor HR at the single-cell level while taking into account chromosome conformation by following global chromatin dynamics upon a single DSB induction. We show that the global increase in mobility induced by a DSB is influenced by the conformation of the chromosome. In the pericentromeric region, a DSB close to the centromere induces a global mobility that we call 'proximal'. This mobility that occurs in trans toward the DSB is dependent on γ-H2A(X), but not on the Rad9-dependent checkpoint nor on the Rad51 nucleofilament. Therefore, this 'proximal' mobility has little influence on the overall HR efficiency, but rather accelerates repair kinetics. In contrast, a subtelomeric DSB induces a mobility that we call 'distal'. This distal mobility only occurs if a homologous sequence is present and if it is near the centromere. We demonstrate that the donor homologous sequence allows γ-H2A(X) spreading that is the requisite for chromatin motion. This distal mobility shares the characteristics of homology search mobility. It is indeed dependent on Rad9 and the Rad51 nucleofilament, and is necessary for HR. Our data unambiguously demonstrate the role of damage-induced global mobility due to non-random chromosome organization and unify conflicting data on its role in HR repair.

## Results

### THRIV, a system to track homologous recombination in vivo

We have recently shown that chromatin mobility favors NHEJ repair in the absence of a donor sequence (*García Fernández et al., 2021*). Here, we asked whether global mobility has a role on DSB repair when a homologous donor sequence is present. To answer this question, we designed a system that simultaneously tracks HR at the single-cell level and measures global genome mobility away from the DSB. We chose to follow HR through fluorescent protein *mCherry* synthesis, using its coding sequence as both recipient and donor sequences. The recipient and donor *mCherry* sequences cannot be translated into functional fluorescent proteins in the absence of HR. The *mCherry* recipient sequence is located on a chromosome. It is under the control of the TEF constitutive promoter and contains an I-*Sce*I cut site (cs), followed by a stop codon that results in an mCherry truncation. The donor sequence, is on a centromeric plasmid, hereafter referred to as dCen. It contains an *mCherry* sequence lacking the promoter and terminator sequences, and shares 539 and 172 bp of homology upstream and downstream of the I-*Sce*I cs of the recipient sequence (*Figure 1A*). The *mCherry* recipient sequence was integrated at three different positions: 5 kb from the centromere of chromosome IV, generating the so-called C strain; in a luminal position, 400 kb from the centromere and 1060 kb from the telomere (L strain), and in a subtelomeric position, 10 kb from the right telomere (S strain) (*Figure 1B*). All strains have similar doubling time (*Figure 1—figure supplement 1A*). These different recipient sequence positions, near and far from the characteristic anchoring features of yeast chromosomes, allow us to investigate distinct chromosome architecture contexts.

Expression of the I-*Sce*I endonuclease on a plasmid under the control of the *GAL1* galactose-inducible promoter generates single DSB (+DSB), whereas the empty plasmid not expressing I-*Sce*I was used as a control (-DSB). I-*Sce*I expression, measured by RT-qPCR showed a 6.7±0.85-fold increase after 1 hr and a fivefold plateau at longer times (*Figure 1—figure supplement 1B*). The cutting efficiency, measured by qPCRs with primers surrounding the I-*Sce*I cs, was 36%±7.8 at 1 hr and reached 77%±0.8 6 hr after I-*Sce*I induction (*Figure 1C*), indicating a delay between the enzyme expression and the effective cleavage. This delay is not observed in the W303 background (*Batté et al., 2017*) and could be due to the BY background used here.

Increasing I-*Sce*I induction time triggered cell cycle arrest and led to a low number of colonies visible by serial dilutions. This effect was seen in all three strains and is thus independent of the chromosomal position of the DSB (*Figure 1D and E*). The presence of dCen plasmid resulted in both, the resolution of 12 hr cell cycle arrest and recovery of survival after DSB induction, as shown by FACS analyses and serial dilutions (*Figure 1D and E*). Expression of mCherry was confirmed by fluorescent microscopy, suggesting that repair occurred by HR (*Figure 1—figure supplement 1C*). The absence of I-*Sce*I cleavage in vitro in DNA extracted from red cells further established HR (*Figure 1—figure supplement 1C*). Red cells were detected by FACS 4 hr after I-*Sce*I induction and their number,

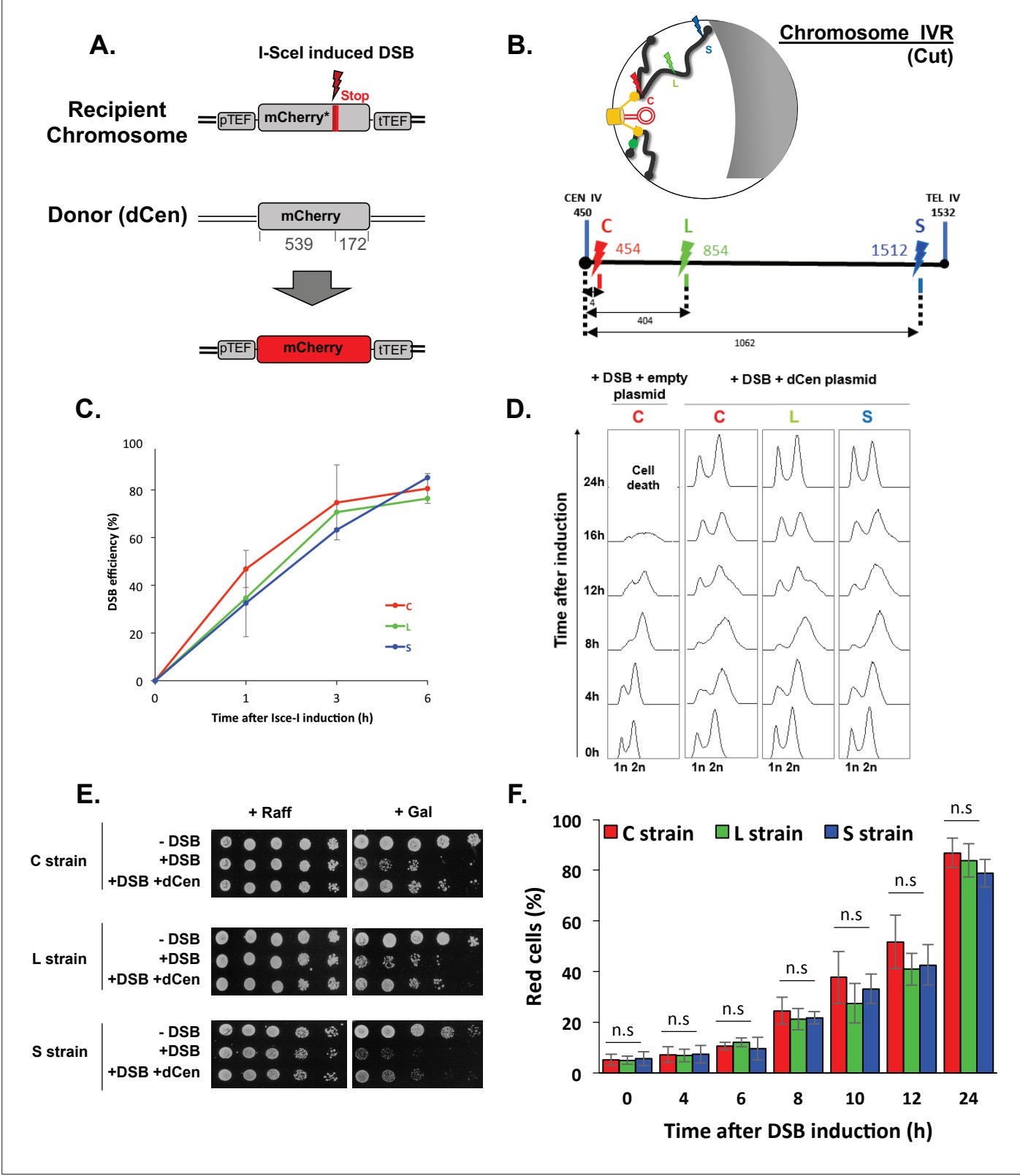

**Figure 1.** Tracks homologous recombination in vivo (THRIV), a system to follow homologous recombination (HR) in vivo. (**A**) Schematic representation of the two sequences used to track recombination events during time. The recipient *mCherry-I-SceI* has a 30pb I-*Sce*I sequence followed by a stop codon (red bar) inserted at 57 amino acids from the end of the *mCherry* coding sequence. The donor (dCen) is a full-length *mCherry* lacking promoter and terminator sequences, carried on a centromeric plasmid. A functional mCherry could be expressed only after HR. (**B**) Schematic 2D representation of the

*Figure 1 continued on next page*

*Figure 1 continued*

Rabl spatial configuration of two chromosomes in the nucleus of *Saccharomyces cerevisiae*. The spindle pole body (rectangle), the microtubules (lines), and the centromeres (spheres) are in yellow. Telomeres are shown as gray/black spheres attached to the nuclear envelope. An orange circle represents the dCen plasmid. The gray crescent symbolizes the nucleolus. Lightening symbolize the targeted I-*Sce*I cutting sites (cs) on chromosome IVR arm, with same color code as in the linear representation represented below. The cs are respectively located at 5 kb from centromere CEN IV (red, strain C), 400 kb from CEN IV (green, strain L), and 20 kb from telomere TEL IVR (blue, strain S), distances are in kb The green dot visualizes the locus used to monitor mobility on undamaged chromosome VR. (**C**) I-*Sce*I cleavage efficiency is measured in a non-donor strain by qPCR using primers flanking the I-*Sce*I cs. The error bars represent the standard deviation of five or two independent experiments for strain C and strains L, S, respectively. (**D**) FACS analysis of the cell cycle of strains C, L, and S after induction of I-*Sce*I with or without dCen plasmid. For the quantification of the cell cycle progression, the amount of DNA is measured with Sytox after fixation with ethanol and treatment with RNase. The peaks corresponding to 1n and 2n amount of DNA are indicated. (**E**) Drop assay (10-fold serial dilutions) showing the sensitivity of C, L, and S strains, containing a plasmid expressing or not I-*Sce*I (-DSB, +DSB, respectively), with dCen plasmid or without (empty plasmid), when I-*Sce*I is induced for 48 hr (galactose) or not (raffinose). (**F**) HR kinetics upon induction of I-*Sce*I in the presence of dCen in C, L, or S strains. Percentages of repaired red cells were measured by FACS after PFA fixation in the absence of Sytox. Error bars represent the standard deviation of three independent experiments, five independent experiments for strain C. Mann-Whitney test, n.s., non-significant.

The online version of this article includes the following source data and figure supplement(s) for figure 1:

**Source data 1.** qPCR values for I-*Sce*I efficiency.

**Source data 2.** FACS values for cell cycle analysis.

**Source data 3.** Spot assay for cell survival to I-*Sce*I induction.

**Source data 4.** FACS values for repaired red cells.

**Figure supplement 1.** Validation of the THRIV system.

**Figure supplement 1—source data 1.** DO values for growth curve.

**Figure supplement 1—source data 2.** Transcription quantitative reverse PCR (RT-qPCR) values for I-*Sce*I expression.

**Figure supplement 1—source data 3.** Red cells values after I-*Sce*I induction.

**Figure supplement 1—source data 4.** Agarose gel after PCR amplification of the I-*Sce*I cutting site and cut by I-*Sce*I in vitro.

**Figure supplement 1—source data 5.** 2D distances values.

which included repaired cells and their progeny, progressively increased with time to 87.08±4.43%, 77.40±0.54%, and 72.58±0.42% at 24 hr in C, L, and S strains, respectively (*Figure 1F*). The lower number of red cells and the lower survival seen in serial dilutions of the S strain could be explained by the greater distance between the DSB near the telomere and the donor sequence. Indeed, the nuclear localization of the dCen plasmid is comparable to that of a pericentromeric sequence (*Figure 1—figure supplement 1D*). Thus, after a single DSB, the system that tracks homologous recombination in vivo (THRIV) is efficient and depends on the relative spatial position between the donor and recipient sequences.

## Global chromatin mobility is enhanced during HR

A positive correlation between global mobility and the ability to repair a DSB by HR has been established (*Challa et al., 2021*; *Cheblal et al., 2020*; *Dion et al., 2012*; *Hauer et al., 2017*; *Miné-Hattab et al., 2017*; *Miné-Hattab and Rothstein, 2012*), but it has also been shown that global mobility does not necessarily have a role in HR efficiency (*Strecker et al., 2016*). To directly investigate global mobility during HR, we tracked TetO repeats inserted at 96 kb from CenV and bound by the tetR-GFP fusion protein named V-Vis, for *vi*sualization of the locus on chromosome V (*Figure 2A* and *Strecker et al., 2016*), in these same strains harboring the THRIV system, that is, I-*Sce*I CS at different positions and the donor sequence on a plasmid. We analyzed the mean squared displacements (MSDs) of V-Vis upon induction by I-*Sce*I in the different C, L, and S strains, in the presence of the plasmids containing or not the donor sequence. For each condition, we tracked the locus by time-lapse microscopy at 100 ms time intervals in 500–1000 cells, as in *Herbert et al., 2017*, and observed mobility for short (10 s) and long (180 s) time scales (*Figure 2*, *Supplementary file 1* and *Figure 2—figure supplement 1A*). Strains expressing or not I-*Sce*I were grown in galactose for 6 hr. In the absence of I-*Sce*I induction, the V-Vis locus exhibited a subdiffusive behavior, with MSDs exhibiting a power law (MSD ~ $Dt^\alpha$), where $\alpha$ was ~0.6. The value of this anomalous exponent below 1 was previously shown at these short and long time scales for many loci scattered along the genome (*Figure 2B*, -DSB, *supplementary file 1* and *Figure 2—figure supplement*

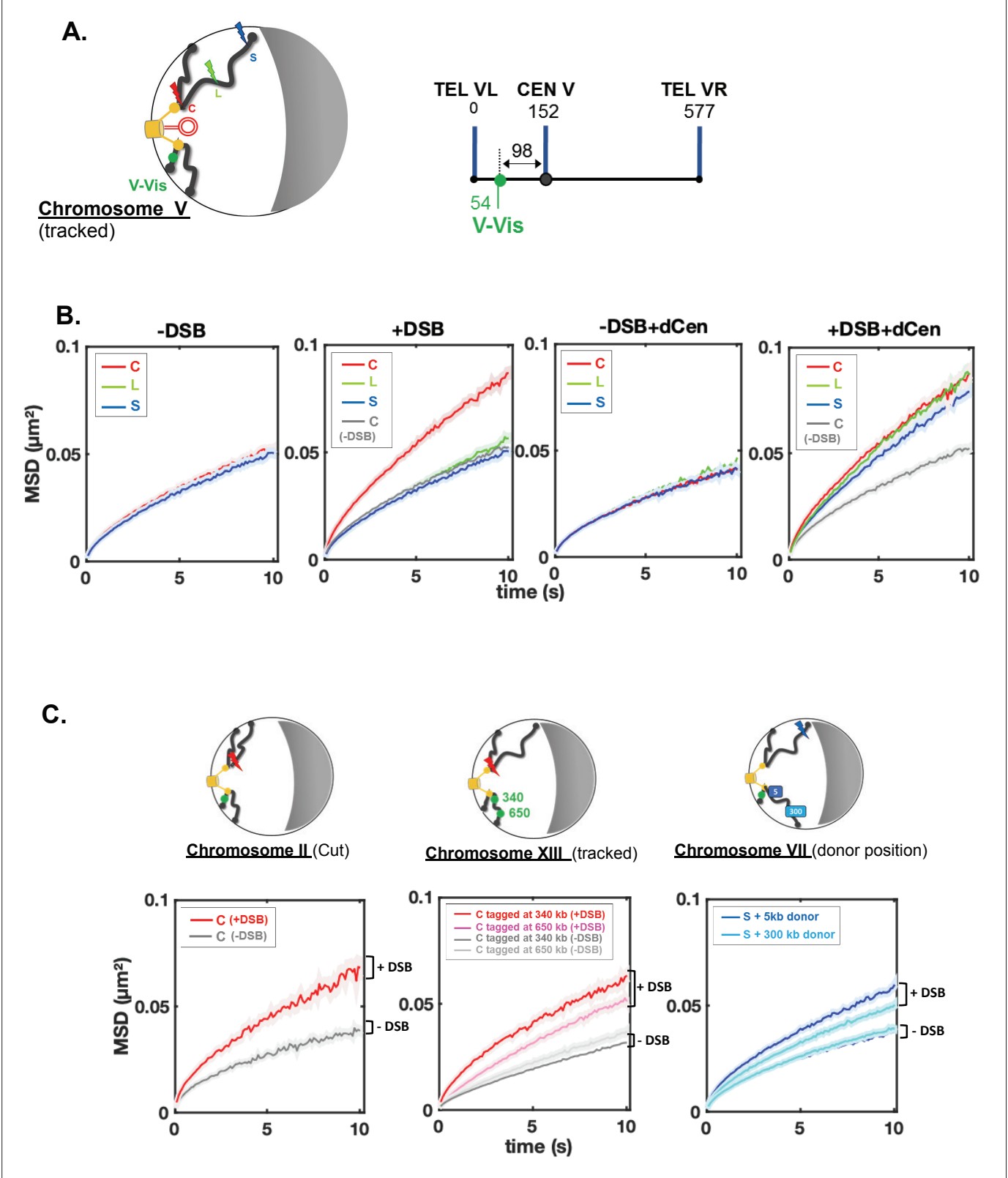

**Figure 2.** Enhanced global chromatin dynamics after damage is dictated by double-strand break (DSB) and donor positions. (**A**) As in *Figures 1B and 2D* and linear schematics of the undamaged chromosome containing the visualization of the locus on chromosome V (V-Vis) locus used for global mobility tracking, indicated as (tracked). V-Vis consists of TetO repeats bound by the TetR repressor protein fused to the fluorescent protein GFP, integrated into the *MAK10* locus on the left arm of chromosome V at 98 kb from the centromere CENV (green dot). (**B**) Mean squared displacements

*Figure 2 continued on next page*

*Figure 2 continued*

(MSDs) as a function of time for the V-Vis locus in strain C (red), L (green), and S (blue) respectively after 6 hr in galactose medium in the absence or the presence of I-*Sce*I expressing plasmid (-DSB, +DSB) and in the presence of the donor plasmid (-DSB+dCen, +DSB+dCen plasmid), or the empty plasmid (-DSB, +DSB). MSDs are calculated from video microscopy data with a time sampling of Δt=100 ms. Between n=405 and n=1794 cells were analyzed for each strain in six independent experiments. The exact number of cells analyzed to calculate each curve (**n**) is indicated in ***supplementary file 1***. The gray curve corresponds to the average MSD of the strain C without DSB in the presence of dCen or of the empty plasmid. (**C**) The global increase in dynamics after 6 hr in galactose medium is verified for (i) another pericentromeric DSB (left); (ii) other tracked loci (middle), and (iii) donor sequences distinct than dCen plasmid (right). (**i**) I-*Sce*I cut site (cs) is inserted at 8 kb from CENII (red curve; n=209), MSD without DSB induction is shown in gray (n=356). Four independent experiments were done. (ii) MSDs are calculated by tracking TetO repeats inserted on chromosome XIII at 340 and 650 kb of the centromere (red and pink curves, n=594 and n=1085, respectively) upon induction of I-*Sce*I DSB on peri-centromere of chromosome IV. The gray curves correspond to the average MSDs without induction for both strains (n=2582 and n=1076, respectively). Seven independent experiments were done; (iii) the donor sequence is inserted on chromosome VII (dChr.VII) after a DSB in subtelomeric position (S, blue lightening). MSDs are measured after insertion of donor sequence at 5 or 300 kb of CEN VII (dark and light blue, n=1403 and n=1496, respectively). -DSB curves represent MSDs when DSB is not induced (n=900 and n=895, respectively). Seven independent experiments were done. 2D schemes of Rabl chromosome conformation with positions of the DSB, the tracked locus, or the donor sequences are shown on the top of each MSD graph as in *Figure 1B*.

The online version of this article includes the following source data and figure supplement(s) for figure 2:

**Source data 1.** Mean squared displacement (MSD) values for *Figure 2B*.

**Source data 2.** Mean squared displacement (MSD) values for *Figure 2C*.

**Figure supplement 1.** Mean square displacements (MSD) at 10s of V-Vis after a single DSB.

**Figure supplement 1—source data 1.** Mean squared displacement (MSD) values.

**Figure supplement 1—source data 2.** Values of mean squared displacement (MSD) at 10 s for different time points of I-*Sce*I induction.

**Figure supplement 1—source data 3.** Values of mean squared displacement (MSD) at 10 s.

*1A*; *Dion et al., 2012*; *Hajjoul et al., 2013*; *Herbert et al., 2017*; *Miné-Hattab and Rothstein, 2012*; *Seeber et al., 2013*; *Spichal et al., 2016*; *Strecker et al., 2016*). Upon I-*Sce*I induction and generation of a single DSB in the C strain, the V-Vis locus showed increased dynamic at the short time scale as well as an increased anomalous α, indicating changes in chromatin properties around the tracked locus (*Figure 2B*, **+**DSB, *supplementary file 1* and *Figure 2—figure supplement 1B*). Global dynamics after 6 hr induction of I-*Sce*I was also observed in strain C at the longer time scale of 180 ms (*Figure 2—figure supplement 1A*) and gradually increased to 1.95-fold after 12 hr (*Figure 2—figure supplement 1B*). In contrast, a single DSB (verified by qPCR, *Figure 1C*) in L and S strains had no effect on V-Vis dynamics, which remained similar to that observed in the absence of DSB induction (*Figure 2B*, +DSB).

Global mobility induced by a pericentromeric DSB is not specific to chromosome IV, since I-*Sce*I cs inserted at 7 kb from CENII induced a similar increase in MSDs of the V-Vis locus (*Figure 2C*, chromosome II, cut). Likewise, the increase in global mobility after a DSB in the C strain was observed elsewhere than at the V-Vis locus, on chromosome XIII at 340 and 650 kb from CENXIII, but to a lesser extent for the locus tracked away from the centromere (*Figure 2C*, chromosome XIII, tracked). Altogether these results show that in the absence of homology, the spatial position of the DSB plays a key role in induction of global mobility. A pericentromeric DSB specifically triggers a global mobility that reflects chromatin changes at short times and greater exploration of nuclear space at long times (*Miné-Hattab et al., 2017*). Furthermore, the gradual increase in mobility over the induction time suggests a progressive chromatin change behind global mobility at short times.

We then analyzed whether global mobility was affected during HR, by measuring global dynamics in the presence of the dCen donor sequence. Before induction, dCen did not alter V-Vis mobility in any of the strains (*Figure 2B*, -DSB+dCen plasmid). Upon DSB, in the presence of dCen, an increase of V-Vis MSDs was now observed even in the L and S strains (*Figure 2B* and *Figure 2—figure supplement 1C*, +DSB+dCen plasmid). We asked whether mobility was also enhanced with an intra-chromosomal donor located on chromosome VII, near and far from the centromere (5 and 300 kb from CENVII). Upon induction of the DSB in the S strain, the closer the donor was to the centromere, the greater the V-Vis mobility was (*Figure 2C*, chromosome VII, donor position). These observations highlight that in addition to chromosome architecture, sequence homology is instrumental in inducing global chromatin mobility. Global mobility may be triggered by homology search only when the DSB is distant from the pericentromeric domain.

## Enhanced chromatin mobility during HR is not dependent on cell cycle arrest but on DNA damage signaling

If the increase in global mobility observed in the presence of a donor sequence can be explained by homology search, then the dynamic behavior of HR-repaired cells should be different from that of cells in search for homology. We therefore distinguished trajectories in the repaired and non-repaired cell populations. Whereas HR-repaired cells were red and cycling, white cells appeared blocked in G2/M, suggesting that these cells had not been repaired by NHEJ and were still searching for homology (*Figure 3A*). MSD at 10 s of V-Vis in white cells was higher than that of red cells (0.088 and 0.064, respectively), while cells without DSBs showed limited global mobility (MSD at 10 s of 0.049) (*Figure 3A* and *Figure 3B*).

Since we do not directly select damaged cells (for instance by analyzing only cells showing a Rad52 focus; *Dion et al., 2012*), we checked for any bias in our analysis. To do so, we considered the two populations of cells where damage is best controlled, that is, (i) the red population that we know has been repaired 12 hr after DSB induction and, more importantly, has lost the cutting site and will not be cut again (undamaged population only) and (ii) the white population, which is blocked in G2/M because it is damaged and not repaired (damaged population only) (*Figure 3A*). These two populations show very significant differences in their median MSDs (*Figure 3—figure supplement 1A*). We artificially mixed the MSD values obtained from these two populations with 20% undamaged and 80% damaged cells. We observed that the mean MSDs of 'damaged-only cells' and 'undamaged-only cells' were significantly different. However, the mean MSD of the damaged-only cells was not statistically different from the mean MSD of the mixed cell population. Thus, the conclusions based on the mean MSD of all cells remained consistent.

To further determine whether the increased dynamics of unrepaired cells is due to their cell cycle arrest in G2/M, we measured V-Vis mobility in a *rad9Δ* checkpoint-deficient mutant. We first verified that the I-*Sce*I cleavage efficiency was similar between wild-type (WT) and *Δrad9* strains and assessed cell survival and cell cycle progression in the *Δrad9* mutant (*Figure 3—figure supplement 1B*). As expected, *RAD9* deletion prevented cell cycle arrest and led to drastically altered cell growth after I-*Sce*I induction (*Figure 3—figure supplement 1C*). Surprisingly, the increase in global chromatin mobility upon DSB induction at pericentromeric site was similar between WT and *Δrad9* cells, regardless of whether dCen was present or not (*Figure 3C*). In contrast, no increase in global dynamics could be observed in the *Δrad9* mutant when DSB was induced away from the centromere, despite effective I-*Sce*I cleavage (*Figure 3C*, *Figure 3—figure supplement 1B*). In the latter context, Rad9 is essential (*Figure 3—figure supplement 1C*), as observed when multiple damages were produced by Zeocin (*García Fernández et al., 2021*; *Seeber et al., 2013*). This indicates that increase in mobility in white cells is not linked to G2/M cell cycle arrest and that the DDR signaling is essential for global mobility only when long-range homology search is required.

Because Rad9 defect prevents the cell cycle arrest required for repair, induction of a DSB in either the C or S strains resulted in cell death in the absence of a donor (*Figure 3—figure supplement 1B*). Furthermore, induction of a DSB resulted in poor growth even in the presence of the donor sequence in the S strain (*Figure 3—figure supplement 1*). In contrast, HR repair of the DSB generated in the pericentromeric region in the C strain was effective and 40% red cells were obtained after 24 hr of induction (*Figure 3D*).

These results show that a G2/M cell cycle arrest is not necessary for the increase in global dynamics and HR repair. Furthermore, the role of Rad9 signaling in global chromatin dynamics is specific to the position of the DSB. Indeed, only the mechanism underlying pericentromeric proximal mobility could effectively proceed without Rad9.

## Enhanced global mobility is controlled by H2A phosphorylation

Phosphorylation of H2A occurs upstream of Rad9 binding after DNA damage. It is necessary and sufficient to induce global mobility when multiple damage occur and it spreads through the centromeres after a pericentromeric DSB (*García Fernández et al., 2021*; *Lee et al., 2014*; *Li et al., 2020*; *Renkawitz et al., 2013*). H2A phosphorylation was thus an attractive candidate in explaining the mechanism by which global mobility occurred in a pericentromeric position-dependent manner. We measured the distribution of γ-H2A(X) by ChIP in *cis* and *trans* of the DSB before and after DSB induction, in the C and S strains (*Figure 4*). ChIP γ-H2A(X) signals in *cis* were detected by qPCR using a set

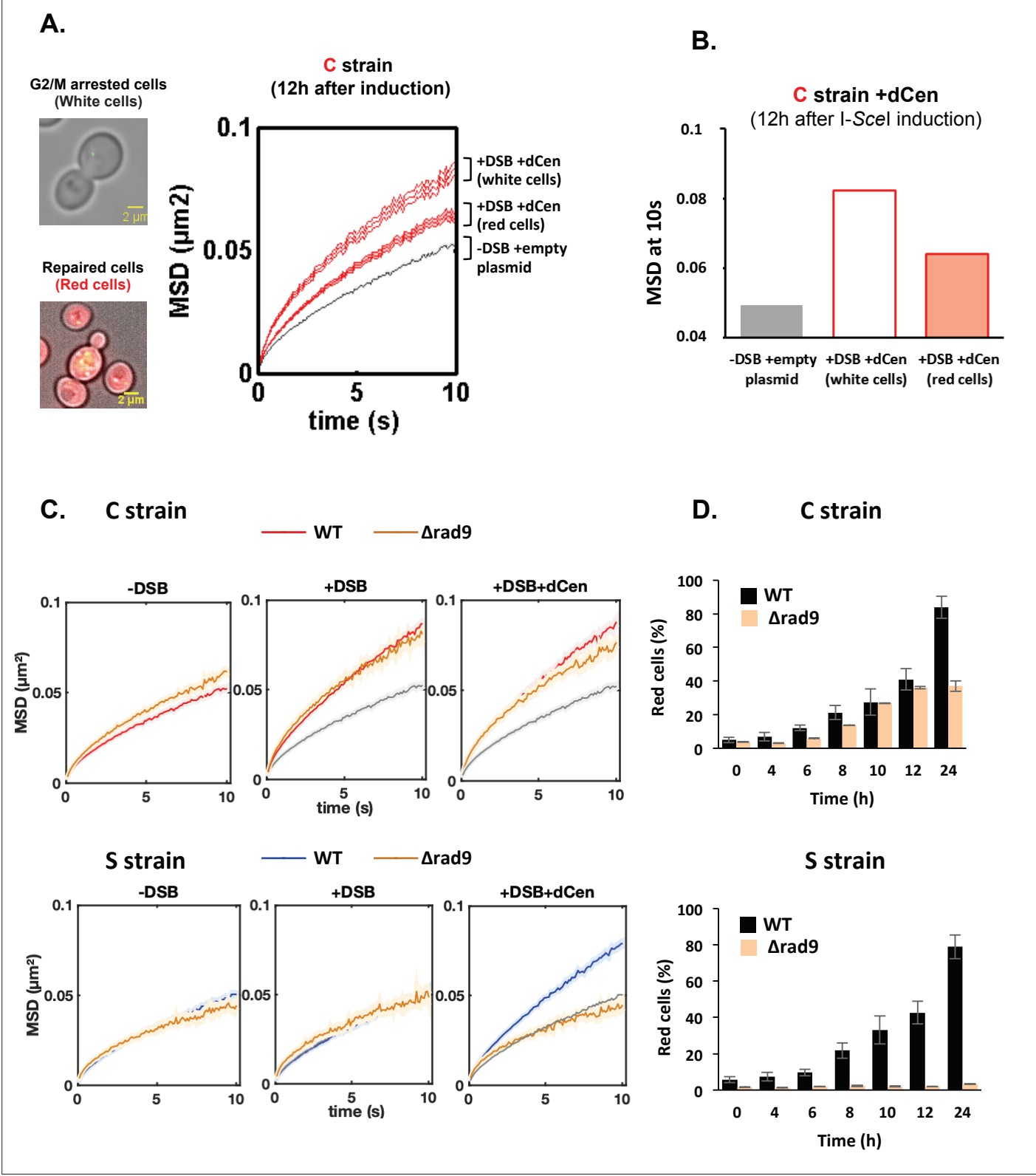

**Figure 3.** Proximal mobility is Rad9 independent. (**A**) Non-repaired cells (white cells, n=584) show increased global mobility, while repaired cells (red cells, n=519) recover normal global mobility. Mean squared displacements (MSDs) of the visualization of the locus on chromosome V (V-Vis) locus in strain C as a function of time, calculated as in *Figure 2A*. The gray and red curves correspond respectively to strain C in the presence of the donor dCen plasmid after 12 hr in galactose medium with a plasmid not expressing I-SceI (-DSB) and expressing I-SceI (+DSB). MSDs are calculated after color visual

*Figure 3 continued on next page*

*Figure 3 continued*

cell sorting. Red MSD full curve corresponds to cycling red cells, red empty curve corresponds to white G2/M arrested cells. Examples of white, G2/M arrested and red, cycling cells are shown. Bar scale, 2 μm. Seven independent experiments were done. (**B**) Box plots of MSDs at 10 s of undamaged, white and red cells 12 hr after I-*Sce*I induction. The color code corresponds to that used in (**A**). (**C**) MSDs in strains C and S in wild-type (WT) and *Δrad9* mutant. MSDs are calculated as in *Figure 2* upon 6 hr of double-strand break (DSB) induction with empty plasmid or with dCen plasmid (+DSB: n=380 [C strain], n=103 [S strain], and +DSB+dCen: n=452 [C strain], n=98 [S strain], respectively). Controls without DSB are also shown (-DSB: n=590 [C strain], n=176 [S strain]). (**D**) Homologous recombination (HR) kinetics upon induction of I-*Sce*I in the presence of dCen in WT (black) and *Δrad9* (orange) C or S strains were measured by FACS as in *Figure 1F*. Error bars represent the standard deviation of three independent experiments.

The online version of this article includes the following source data and figure supplement(s) for figure 3:

**Source data 1.** Mean squared displacement (MSD) values.

**Source data 2.** Mean squared displacement (MSD) values at 10 s.

**Source data 3.** Mean squared displacement (MSD) values.

**Source data 4.** Values for repaired red cells.

**Figure supplement 1.** Implication of Rad9 in survival, cell cycle and global dynamics.

**Figure supplement 1—source data 1.** qPCR values for I-*Sce*I efficiency.

**Figure supplement 1—source data 2.** Spot assay for cell survival to I-*Sce*I induction.

**Figure supplement 1—source data 3.** FACS values for cell cycle analysis.

**Figure supplement 1—source data 4.** Mean squared displacement (MSD) values at 10 s.

of primers located 5, 50, and 100 kb from the I-*Sce*I cutting site. γ-H2A(X) signal on the actin gene was used as a reference control. In the absence of DSB, low-intensity γ-H2A(X) ChIP signals were detected in both the C and S strains (*Figure 4A*, gray bars). After 6 hr of I-*Sce*I induction, ChIP levels of γ-H2A(X) increased significantly in *cis* at 50 and 100 kb from the I-*Sce*I CS by approximately 15- and 10-fold in the C and S strains, respectively (*Figure 4A*, white bars). No increase in ChIP γ-H2A(X) levels was detected at 5 kb from the CS, likely due to 5′ to 3′ resection occurring from the DSB ends (*Eapen et al., 2012*; *Renkawitz et al., 2013*). In the presence of the donor, the enrichment of γ-H2A(X) was comparable to that observed with the empty plasmid, except at 100 kb from the cutting site, where a decrease was observed, probably indicating repair (*Figure 4A*, red bars). To next examine the levels of γ-H2A(X) relative to the centromere in *trans*, specific primers were designed at 15, 50, and 100 kb in the region between the CENV and the V-Vis locus (*Figure 4B*). In the absence of the donor, enrichment of γ-H2A(X) was observed only if DSB was induced near the centromere (C strain). ChIP levels of γ-H2A(X) increased sixfold in *trans* at 15 kb and twofold at 50 and 100 kb from CEN-V (*Figure 4B*, white bars). This is consistent with γ-H2A(X) propagation occurring across centromeres that are spatially close. Notably, the V-Vis locus, located 90 kb from CENV, corresponds to a position where γ-H2A(X) ChIP signals were increased twofold, suggesting a correlation between increased V-Vis mobility and higher γ-H2A(X) levels. Strikingly, an enrichment of γ-H2A(X) ChIP levels was found in *trans* with a similar increase in both the C and S strains, in the presence of dCen, but not of the empty plasmid (*Figure 4B*). Together, these results reveal that the presence of a donor sequence spatially close to the centromere is critical for the propagation of γ-H2A(X) in *trans*, independently of the genomic position of the DSB. They further support that γ-H2A(X) spreads around the break and across pericentromeric regions if the DSB is close to the centromere.

Given the close correlation between H2A phosphorylation around V-Vis and the increased mobility at this locus, we further explored the involvement of γ-H2A(X) in mobility by measuring global dynamics in phospho-deficient H2A-S129A and phospho-mimetic H2A-S129E mutants, in the presence or absence of dCen. I-*Sce*I cleavage efficiency in these mutated strains exhibited similar levels to the WT after 6 hr of I-*Sce*I expression, as shown by qPCR (*Figure 4—figure supplement 1A*). Upon I-*Sce*I induction, serial dilutions of cells showed that colony formation was impaired in the C and S strains in the absence of dCen in either a WT or H2A-S129A mutant background (*Figure 4—figure supplement 1B*). Yet, the H2A-S129E mutant showed better survival, as expected by its positive effect on NHEJ repair (*García Fernández et al., 2021*). In the presence of dCen, colonies could be formed in all strains. However, there were fewer colonies in the S strain mutated for H2A-S129A, indicating that H2A phosphorylation is required for cell survival mainly when the DSB is distant from the centromere.

We next analyzed global mobility upon DSB induction in WT and H2A-S129 mutant strains. In the H2A-S129A mutant, the increase in global chromatin mobility was similarly impaired in the C and S

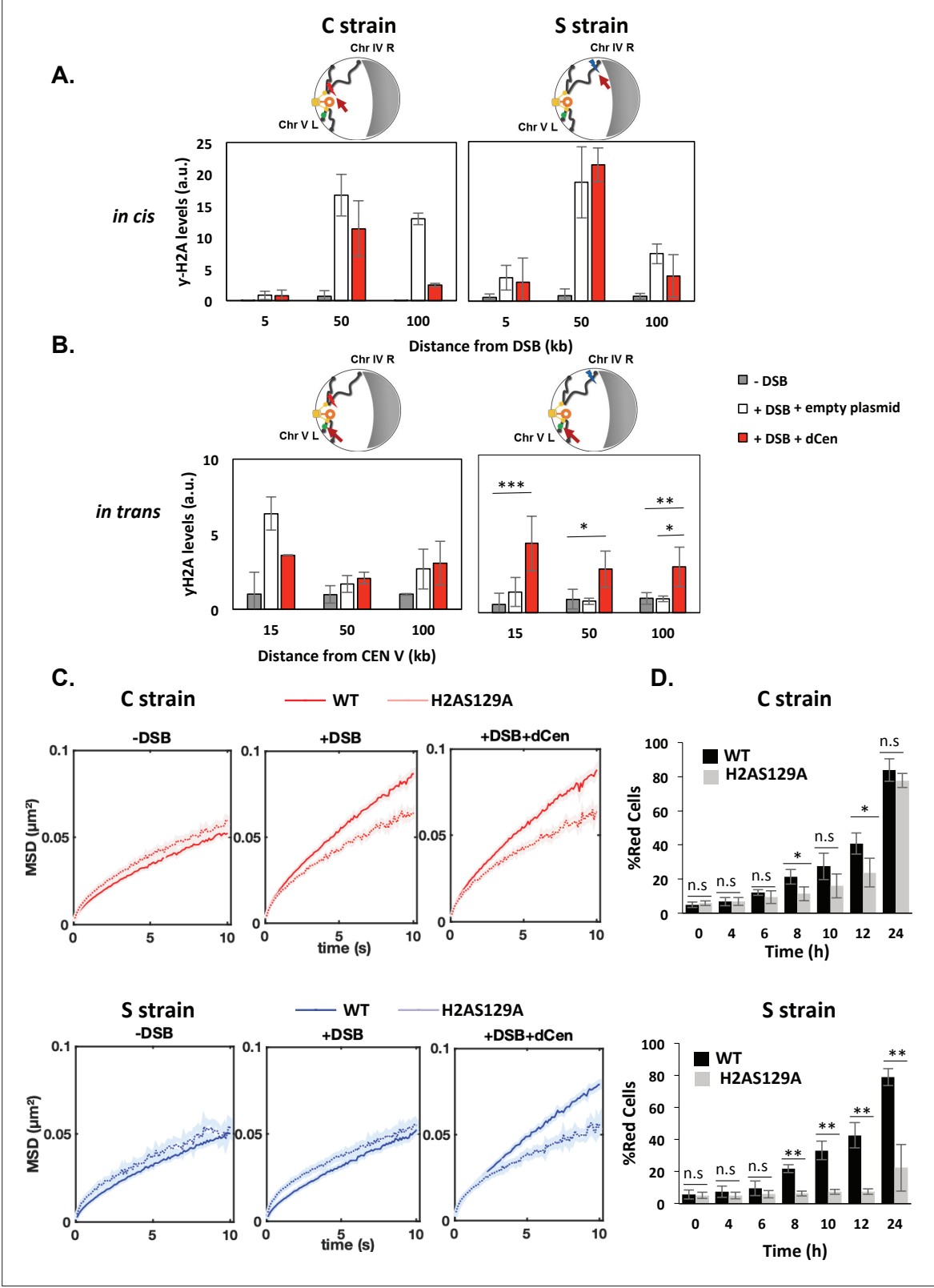

**Figure 4.** Double-strand break (DSB) induction in the presence of dCen triggers γ-H2A(X) spreading in *trans*. (**A**) H2A phosphorylation spreading measured by chromatin immunoprecipitation (ChIP) around I-*Sce*I cutting site (cs) in strain C or S (in *cis*) in the absence (gray bars) or in the presence of DSB in the presence of the empty plasmid (white bars) or the dCen plasmid (red bars). ChIP was performed with γ-H2A(X) antibody after 6 hr induction of I-*Sce*I. Uncut control is shown as empty bars. DNA was analyzed by quantitative PCR using 'in *cis*' primers corresponding to sequences

*Figure 4 continued on next page*

*Figure 4 continued*

at 5, 50, and 100 kb from the right and left sides of I-*Sce*I cs in strains C and S, respectively. Actin was used as the reference gene for each condition. Each bar represents the γ-H2A(X) fold enrichment (γ-H2A(X)-IP/input relative to actin-IP/input) for undamaged and damaged conditions, respectively. The error bars represent the standard deviation of three independent experiments. (**B**) γ-H2A(X) spreading 'in *trans*' around pericentromeric region of chromosome V. ChIP was performed as above using 'in *trans*' primers corresponding to sequences at 15, 50, and 100 kb from CENV. Note that *v*isualization of the locus on chromosome V (V-Vis) is positioned at 95.7 kb from CENV. (**C**) Mean squared displacements (MSDs) in strains C and S in H2A-S129A mutated backgrounds. MSDs are calculated as in *Figure 2* upon 6 hr of DSB induction with empty plasmid or dCen plasmid (+DSB: n=308 [C strain], n=192 [S strain] and +DSB+dCen: n=205 [C strain], n=198 [S strain], respectively). Controls without DSB are also shown (-DSB: n=352 [C strain], n=88 [S strain]). Five independent experiments were done. Wilcoxon rank-sum test between distributions, with the p-value. n.s., non significant, * (p<0.05), ** (p<0.001). (**D**) Homologous recombination (HR) kinetics upon induction of I-*Sce*I in the presence of dCen in strains C or S in wild-type (WT) (black) and H2A-S129A (light gray) mutant were measured by FACS as in *Figure 1F*. Error bars represent the standard deviation of three independent experiments. Wilcoxon rank-sum test between distributions, with the p-value. n.s., non-significant, * (p<0.05), ** (p<0.001).

The online version of this article includes the following source data and figure supplement(s) for figure 4:

**Source data 1.** qPCR values after chromatin immunoprecipitation (ChIP) at different indicated genomic positions in *cis* of the double-strand break (DSB).

**Source data 2.** qPCR values after chromatin immunoprecipitation (ChIP) at different indicated genomic positions in trans of the double-strand break (DSB).

**Source data 3.** Mean squared displacement (MSD) values.

**Figure supplement 1.** Implication of H2AS129 in survival, cell cycle and global dynamics.

**Figure supplement 1—source data 1.** qPCR values for I-*Sce*I efficiency.

**Figure supplement 1—source data 2.** Spot assay for cell survival to I-*Sce*I induction.

**Figure supplement 1—source data 3.** Mean squared displacement (MSD) values.

**Figure supplement 1—source data 4.** Values for repaired red cells.

**Figure supplement 2.** Mean square displacements (MSD) at 10s of V-Vis after a single DSB in the H2AS129 mutants.

**Figure supplement 2—source data 1.** Mean squared displacement (MSD) values at 10 s.

strains harboring or not the donor sequence dCen (*Figure 4C* and *Figure 4—figure supplement 2A*). In contrast, the MSD curves of the phosphomimetic mutant H2A-S129E showed an increase in both the C and S strains, even in the absence of DNA damage (*Figure 4—figure supplement 1C* and *Figure 4—figure supplement 2A*). These results clearly indicate that global mobility following a single DSB is primarily controlled by H2A(X) phosphorylation, presumably by spreading of this mark. This spreading may be triggered by a DSB near the pericentromeric domain, or by a centromeric donor when the DSB is induced elsewhere in the genome.

The kinetics of HR repair in H2A-S129 mutants was next assessed using the THRIV system (*Figure 4D*). After a DSB generated near the centromere, the kinetics of red cell appearance was reduced in the H2A-S129A mutant compared to the WT, but the number of cells repaired by HR after 24 hr was comparable in both strains (*Figure 4D and C*, gray bars). In contrast, in the same H2A-S129A mutant, a DSB generated in the S strain could not be repaired by HR (*Figure 4D and S*, gray bars). HR kinetics in the S129E mutant was similar to that in WT strains, regardless of the DSB position (*Figure 4—figure supplement 1D*, gray bars). These results show that γ-H2A(X)-mediated mobility upon DSB induction is essential to facilitate HR. However, the need for mobility and 3D spatial sampling are determined by the position of the DSB and the donor sequence in the nucleus.

## Global chromatin mobility requires Rad51 for long-range homology search only

To further investigate whether global mobility is dependent on homology search, we tested the role of the repair protein Rad51 (*Deng et al., 2015*; *Zou and Elledge, 2003*). It has been proposed that the 3' single-stranded DNA extension covered by Rad51 searches for homologous sequences around the DSB and beyond within chromosomes, in a way that depends on their spatial organization (*Renkawitz et al., 2013*). Rad51 has also been implicated in homology search in diploid cells (*Miné-Hattab et al., 2017*; *Miné-Hattab and Rothstein, 2012*; *Smith et al., 2018*). It is further documented that γ-H2A(X) and Rad51 propagate concomitantly (*Renkawitz et al., 2013*). Given our observations, Rad51 might be involved differently depending on the relative positions of the donor and recipient sequences. To test this, we deleted *RAD51* in the C and S strains, verified that cleavage by I-*Sce*I was as effective

in both strains (*Figure 5A*), and measured global dynamics in the presence or absence of damage (*Figure 5B* and *Figure 5C*). In the absence of a DSB, the global dynamics of WT and *Δrad51* strains were similar (*Figure 5B* and *Figure 5C*, -DSB). Upon I-*Sce*I induction, the *Δrad51* mutant strikingly did not affect the global mobility observed in the C strain (*Figure 5B* and *Figure 5C*, C strain, +DSB). In contrast, deletion of *RAD51* abolished the increase in global dynamics induced in the S strain when dCen was present (*Figure 5B* and *Figure 5C*, S strain+DSB+dCen). These results substantiate that global mobility could require Rad51 for long-range homology search and highlight the existence of a Rad51-independent global mobility when the damage is close to the centromere.

## Discussion

Increase of mobility of a genome damaged by DSBs is a universal response. How this occurs and what the consequences are for genomic integrity are puzzling and not yet fully understood. Here, we provide direct evidence that two types of global mobility are involved in HR in a chromosome organization-dependent manner, in part dependent on H2A phosphorylation.

We establish the critical role of chromosome organization in the response to damage by increasing global genome mobility. Hence, the position of the DSB, away from or in proximity to the centromeres, engages more or less complex mechanisms for the establishment of global mobility. This finding is coherent with previously discovered role of yeast Rabl chromosome configuration in DSB repair efficiency (*Agmon et al., 2013*; *Batté et al., 2017*; *Lee et al., 2015*). In this chromosome configuration, all centromeres, anchored at one pole of the nucleus, form a region where *trans* contacts are enriched (*Berger et al., 2008*; *Duan et al., 2010*; *Lazar-Stefanita et al., 2017*). Sequences located in territories whose spatial proximity is defined by this chromosome organization were shown to be repaired by HR more efficiently than spatially distant regions, with a striking efficient HR when DSBs were generated in the chromatin close to the centromeres (*Agmon et al., 2013*; *Batté et al., 2017*; *Lee et al., 2015*). This pericentromeric chromatin, corresponding to a 20–50 kb region flanking the 125 bp yeast point centromere (*Paldi et al., 2020*), is particularly enriched in chromosome structural maintenance complexes (SMCs) that include cohesins, condensins, and Smc5/6. SMCs have a strong impact on chromatin loop formation and size (*Betts Lindroos et al., 2006*; *Dauban et al., 2020*; *Lazar-Stefanita et al., 2017*; *Piazza et al., 2021*; *Weber et al., 2004*), further suggesting a link between the particular structure of pericentromeric chromatin and HR repair efficiency. Interestingly, the recent discovery of the repeated emergence of Rabl-like configurations during evolution may point to a functional advantage to centromeric clustering (*Hoencamp et al., 2021*).

We observed a first type of global mobility, that we propose to name 'proximal mobility', when a DSB is induced near the centromeres. This proximal mobility correlates with and depends solely on γ-H2A(X) that can spread from the site of damage to other pericentromeric regions. This was shown by ChIP in a strain where the break is induced near the centromere (C strain), in agreement with *Lee et al., 2014*; *Li et al., 2020*; *Renkawitz et al., 2013*. Proximal mobility is not altered by the presence of a donor sequence, by checkpoint progression, or by the Rad51 nucleofilament, as evidenced by the lack of effect of *RAD9* (53BP1 ortholog) or *RAD51* deletions on mobility, indicating that at least under these conditions, γ-H2A(X) spreading allowed by centromeres clustering is sufficient to enhance chromatin mobility.

That Rad9 is dispensable for proximal mobility is intriguing and contradicts previous evidence showing a checkpoint requirement for increased global mobility after random damage (*Seeber et al., 2013*; *Smith et al., 2018*). This discrepancy could be explained by the nature of the induced DSBs, as random DSBs triggered by Zeocin or γ-rays could induce high DDR, promoting both γ-H2A(X) and recruitment of the checkpoint machinery throughout the genome (*Seeber et al., 2013*; *Smith et al., 2018*). Furthermore, as proposed by *Hauer et al., 2017*, the checkpoint machinery could trigger recruitment of INO80, a chromatin remodeler complex important in nucleosome stability, linked to global dynamics after the induction of multiple DSBs (*Cheblal et al., 2020*; *Hauer et al., 2017*). Here, the single DSB generated near a centromere induces DDR activation and H2A phosphorylation, but propagation of γ-H2A(X) through centromeric regions is sufficient to induce global mobility. Rad9 could be dispensable because its recruitment remains limited in the regions flanking the DSB (*Figure 5*; *Clouaire et al., 2018*; *Ferrari et al., 2015*; *Renkawitz et al., 2013*).

How does the *trans*-propagation of γ-H2A(X) in the pericentromeric region occur? γ-H2A(X) is primarily induced by the H2A kinase Mec1 (*Lee et al., 2014*; *Li et al., 2020*; *Renkawitz et al., 2013*).

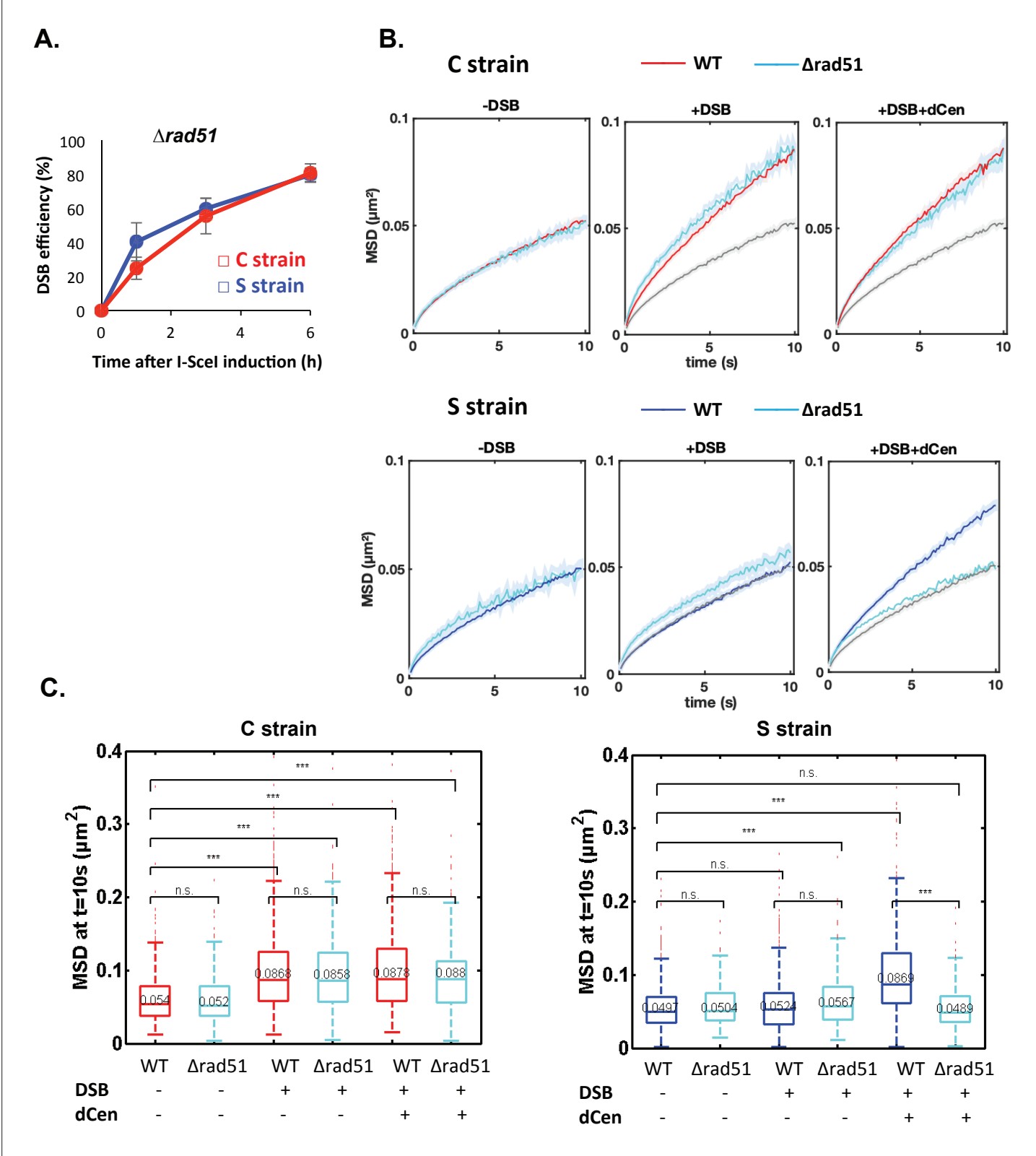

**Figure 5.** Global mobility requires Rad51 only when double-strand break (DSB) and donor sequence are spatially distant. (**A**) I-*Sce*I cleavage efficiency is measured in *Δrad51* strains by qPCR using primers flanking the I-*Sce*I cutting site. The error bars represent the standard deviation of three independent experiments. (**B**) Mean squared displacement (MSD) of the *vi*sualization of the locus on chromosome V (V-Vis) locus in strains C (red) or S (blue) mutated for Rad51 (*Δrad51*) after 6 hr in galactose medium when in the presence of an empty plasmid, I-*Sce*I is not expressed (left, -DSB: n=235 [C

*Figure 5 continued on next page*

*Figure 5 continued*

strain], n=116 [S strain]), I-*Sce*I is expressed (middle, +DSB: n=416 [C strain], n=368 [S strain]), or I-*Sce*I is expressed in the presence of dCen plasmid (right, +DSB+dCen: n=218 [C strain], n=244 [S strain]). The gray curve corresponds to the control without DSB (-DSB: n=235 [C strain], n=116 [S strain]). Five independent experiments were done. (**C**) Boxplots of the distribution of MSDs at 10 s from **B** for strains C and S in wild-type (WT) and *Δrad51* background without or with a DSB (-DSB, +DSB), with dCen or empty plasmids (-dCen, +dCen). Median values, lower and upper quartiles are shown. Whiskers indicate the full range of measured values, except for outliers represented by small red dots. Parentheses indicate the result of a Wilcoxon rank-sum test between distributions, with the p-value. n.s., non significant, * (p<0.05), ** (p<0.001), *** (p<0.0001).

The online version of this article includes the following source data for figure 5:

**Source data 1.** qPCR values for I-*Sce*I efficiency.

**Source data 2.** Mean squared displacement (MSD) values.

**Source data 3.** Mean squared displacement (MSD) values at 10 s.

Our observation that proximal mobility is solely dependent on γ-H2A(X) is consistent with a model of Mec1 diffusion across centromeres. As suggested by a recent study combining experimental γ-H2A(X) ChIP in the Δ*mec1* mutant and mathematical modeling, Mec1 may diffuse in 3D from the site of damage (*Li et al., 2020*). The attenuated diffusion of Mec1 along the chromatin fiber reported by Li et al. could also explain why tracking loci farther from the centromere, 340 or 650 kb from it, reveals a smaller increase in global mobility (*Figure 2C*, *supplementary file 1*). Consistent with the proximal mobility, cohesins or Smc5/6 – which are recruited during DSB repair (*Betts Lindroos et al., 2006*; *Caridi et al., 2018*; *Ström et al., 2004*; *Torres-Rosell et al., 2007*; *Unal et al., 2007*) – could promote H2A phosphorylation during loop extrusion, creating a functional repair unit, as has been proposed for mammalian TADs (*Arnould et al., 2021*). Although the loop model may not match with the bending properties of yeast chromatin (*Li et al., 2020*), the gradual increase in mobility we observed in the C strain is consistent with progressive phosphorylation of H2A along the chromosome over time. It would be interesting to test, using polymer models, how spreading of γ-H2A(X)-induced chromatin modifications allows mobility in chromosomal regions farther from the centromere.

Whatever the mechanism of propagation, phosphorylation could promote mobility through a stiffer chromatin, due to the repulsion of phosphate negative charges, as we have proposed (*García Fernández et al., 2021*; *Herbert et al., 2017*). H2A phosphorylation could also differentially modify chromatin structure by modulating the recruitment of INO80 (*Cheblal et al., 2020*; *Hauer et al., 2017*; *van Attikum et al., 2004*; *Bennett et al., 2013*), and SMCs (*Cheblal et al., 2020*; *Mirny and Dekker, 2021*), or modify the links between microtubules and centromeres, although the loss of centromeric anchoring as a source of global mobility is still debated (*Cheblal et al., 2020*; *Strecker et al., 2016*; *Lawrimore et al., 2017*).

We describe a second type of global mobility, which we propose to call 'distal mobility', when the damage occurred spatially far from the centromeres and donor sequence, such as in the middle or at the end of the IVR chromosome arm (L and S strains). In this case, the increased mobility of V-vis is only induced if a donor close to the centromere, either on a plasmid or on the chromosome, is present. This donor-dependent distal mobility requires a more sophisticated mechanism than simple γ-H2A(X) propagation, since it involves checkpoint progression and homology search and is thus triggered in a Rad9- and Rad51-dependent manner.

Interestingly, these genetic requirements are the same as those described for local DSB mobility, drawing for the first time a link between local and global mobility. Our observations suggest that checkpoint activation, involving Rad9 recruitment and H2A phosphorylation, initiates mobility at the damaged locus (local mobility). Likewise, Rad51 also collaborates to maintain this local mobility, preparing itself to initiate sampling within the nucleus with an elongated filament. In agreement, ChIP of γ-H2A(X), Rad51, and H2A-S129 mutants show that γ-H2A(X) signal propagation and homology search are apparently directly related (*Figure 4* and *Renkawitz et al., 2013*). The fact that the presence of a donor is required to observe both H2A phosphorylation and increased distal mobility seems to confirm genome sampling by the Rad51 nucleofilament together with Mec1 as proposed by *Renkawitz et al., 2014*; *Renkawitz et al., 2013*. Furthermore, it implicates the recombination process itself, as well as the time of residency of the Rad51 nucleofilament, in γ-H2A(X) spreading near the donor. The stability of Rad51 at the donor sequence involves Rad9 (*Ferrari et al., 2020*), further explaining the role of Rad9 in distal mobility. Our observations thus suggest that recombination would be the cause of distal mobility, in contrast to proximal mobility, which may exist independently of

HR. Interestingly, Piazza et al. found that a DSB involves a cohesin-dependent scaffold of the broken chromosome, which isolates that chromosome from others and inhibits *trans* contacts (*Piazza et al., 2021*). The distal mobility we describe here could help overcome this isolation and facilitate *trans* homology searches when needed.

In addition to these genetic requirements shared by the local and global motions induced after DSBs, the question of the involvement of external forces due to nuclear filaments in both motions is raised. The release of centromeres from their attachment to nuclear microtubules emanating from the spindle pole body has been suggested as a source of global mobility (*Lawrimore et al., 2017*; *Strecker et al., 2016*). On the other hand, cytoplasmic microtubule-induced local mobility in *Schizosaccharomyces pombe* appears to be important for HR repair (*Zhurinsky et al., 2019*), and DSB-induced intranuclear microtubules have recently been implicated in directed mobility of the damaged site (*Oshidari et al., 2018*). Beyond yeast, directed mobilization of damaged DNA involving microtubules, nuclear actin, or nuclear motors such as myosin and kinesin has been observed in metazoans, pointing to their evolutionary conservation (*Caridi et al., 2018*; *Lamm et al., 2020*; *Schrank et al., 2018*; *Zhu et al., 2020*). Importantly, these directed motions were involved for the homology-directed repair of DSBs and release of replication stress (*Caridi et al., 2018*; *Laflamme et al., 2019*; *Lamm et al., 2020*; *Schrank et al., 2018*). Although it is not yet known whether nuclear filaments are similarly engaged in local and global movements such as those observed here, their involvement would support the idea that multiple mechanisms are required for DNA damage-induced mobility, the ultimate goal of which is repair and genomic stability.

Of note, the proximal and distal mobility described here have different roles in HR repair, as documented by directly tracking cells repaired by HR in vivo. While distal mobility is a product of HR repair, proximal mobility serves as a dispensable booster (*Figure 6*). Indeed, H2A-S129A mutants are able to form a functional mCherry by HR, and therefore red colonies, when the cut is centromere-proximal (C strain), although proximal mobility is not increased. Red cells appear more slowly than in the WT but reach a comparable level between the two strains after 24 hr. Thus, HR can take place without the increase in mobility, but proximal mobility accelerates it. Furthermore, in the C strain deleted for the *RAD9* checkpoint gene, red cells surprisingly appear, although there are fewer of them than in the WT strain. In the *Δrad9* mutant, proximal mobility is not affected but the cell cycle arrest required for HR is inefficient. This suggests that proximal mobility at least partially suppresses the recombination defects associated with the absence of Rad9. This observation is remarkably consistent with our recent findings establishing that the global mobility observed in the H2A-S129E mutant suppresses the *Δrad9* checkpoint defect (*García Fernández et al., 2021*).

## Conclusion

Chromosome organization influences chromatin mobility when damage occurs, which can accelerate repair either by NHEJ (*García Fernández et al., 2021*; *Ma et al., 2021*) or by HR as shown here. The global chromatin mobility observed in different regions of the nucleus could thus be a way to coordinate different responses to DNA damage. The proximal mobility observed in the pericentromeric region, in particular, may be an additional mean of ensuring the essential maintenance of centromeres (*Yilmaz et al., 2021*). Interestingly, a sophisticated biological function such as sleep has been associated with chromosome mobility (*Zada et al., 2019*). In zebrafish, a link was found between neuronal activity, chromosome mobility, and the number of lesions, also highlighting the protective role of chromosome mobility. Universally, global mobility would be an additional warrant of genomic integrity.

## Limitations of the system

Although the THRIV system used in this study is efficient in inducing DNA damage and monitoring both chromatin mobility and HR repair, it presents certain limitations. First, the continuous induction of the enzyme that leads to a persistent DSB can affect the DDR, the choice of DSB repair pathway, and probably, chromatin behavior (*Shahar et al., 2012*). To bypass this limitation, a 'degron' system would allow inhibiting rapidly the I-*Sce*I expression and thus, measuring dynamics in a context of non-persistent damage. Second, the time of induction of 6 hr is long. Despite the high efficiency of DSB generation after 3 hr of induction, we however decided to measure the mobility 6 hr after DSB induction because this time corresponds to the time required for the γ-H2A *trans* spreading (*Lee et al., 2014*; *Renkawitz et al., 2013*; *Figure 4B*). Third, we did not distinguish between undamaged

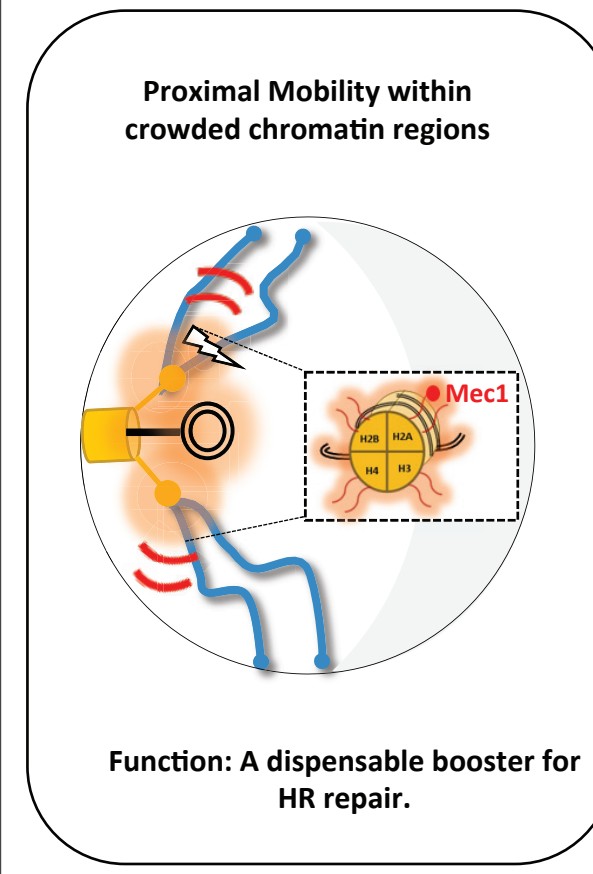

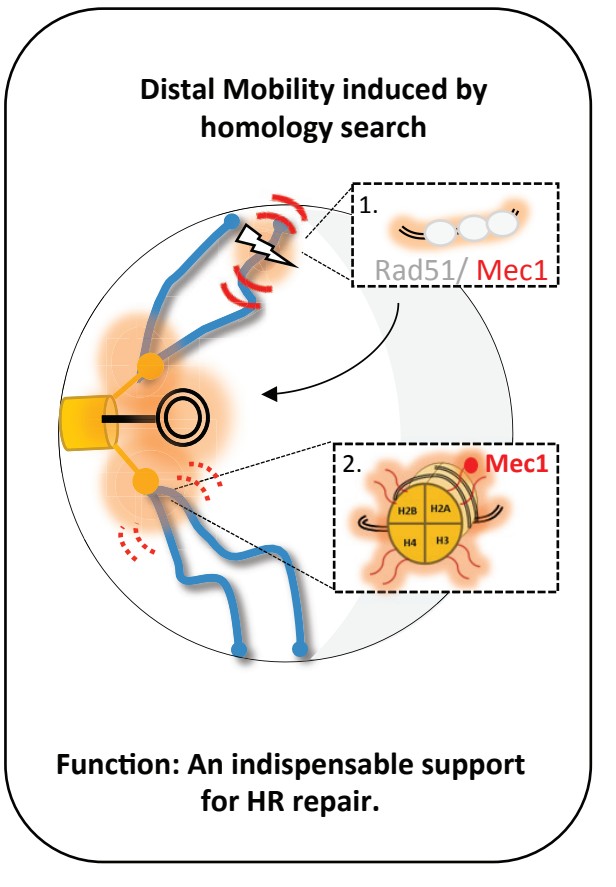

**Figure 6.** Two types of global mobility are involved in double-strand break (DSB) repair by homologous recombination. A proximal mobility (left) occurs in the pericentromeric region, a nuclear domain where *trans*-contacts are enriched. Only H2A phosphorylation is involved in this global mobility, which serves to increase the rate of homologous recombination (HR). By analogy to painting techniques, the proximal mobility could reflect the pouring technique in which oil paint is deposited on water and spreads multidirectionally. It is likely that Mec1 or Tel1 are the factors responsible for this spreading pattern. Distal mobility (right) occurs when DSB is initiated in a region far from the centromeres. The Rad51 nucleofilament is required, as well as HR to promote H2A phosphorylation. The painting analogy here would be that of a paintbrush, with the handle being the Rad51 nucleofilament and the brush being Mec1. The movement of the handle depends on Rad9, and the moving brush allows the deposition of γ-H2A(X). In the absence of Rad51 or Rad9, the brush stays still, decorating only H2A in its immediate environment. The amount of paint deposited will depend on the residence time of the Rad51-ADNsc-Mec1 filament in a given nuclear region. Chromatin decorated by γ-H2A(X) is represented by the orange shadow, at the DSB site and in the pericentromeric region. The red curves represent mobility with thickness symbolizing the amplitude of motion, thin curves for global mobility, and thick curves for local mobility. The lightning bolt represents the DSB.

and damaged cells. Even if we could show that repaired red cells behave differently that unrepaired white cells (*Figure 3*), at a given time, it is possible that a cell, although damaged, moves little and conversely that a cell moves more, even if not damaged, raising the important and still open question of cell to cell heterogeneity (*Altschuler and Wu, 2010*). Fourth, we measured the time-averaged MSDs at a short sampling rate (100 ms), which reliably detects the increase in subdiffusive motion upon DSB induction. Although powerful, this measurement method does not detect subtle changes in mobility such as directed motion that might be masked by overlapping periods of subdiffusive motion (*Oshidari et al., 2018*). How random and directed motions of damaged DNA intersect to help repair deserves to be interrogated. Finally, the THRIV system is based on a plasmid donor. Although the THRIV system has also been validated with a chromosomal donor, the plasmid/chromosome chromatin contexts are different, and may notably have an impact on the residence time of the nucleoprotein filament. This question opens the door to modeling approaches on the role of chromatin context in its dynamics and repair efficiency by HR.

# Materials and methods

## Yeast culture

Yeast cells were grown in rich medium (yeast extract-peptone-dextrose, YPD) or synthetic complete (SC) medium lacking that appropriate amino acid at 30°C. Synthetic medium containing 2% raffinose and lacking appropriate amino acids was used to grow the cells overnight prior the induction of I-SceI by plating onto 2% galactose plates or addition of 2% galactose in liquid culture. For high throughput in vivo spot tracking, cells were plated on agarose patches (made of SC medium containing 2% agarose and galactose) and sealed using VaLaP (1/3 Vaseline, 1/3 Lanoline, and 1/3 Paraffin). For growth curves, overnight cultures were diluted in YPD to a starting density of OD600=0.5. Cell cultures were incubated at 30°C and OD was measured every 45 min. For drop assays, overnight cultures were diluted to bring cells to the exponential phase. These cultures were diluted to a starting density of OD600=1.0 and serial 1:5 dilutions were plated on selective medium (SC-Ade-His) containing raffinose or galactose (2%), and incubated at 30°C for 48 or 72 hr.

## Yeast strains and plasmids

Strains C, L, and S were constructed by first integrating 256 TetO sequences at the MAK10 gene (YEL053C) as in *Strecker et al., 2016*. To insert I-SceI cutting site at the desired position, a *KANMX* cassette was integrated into intergenic loci of chromosome IV at positions 453880+453959 (C strain); 853996+854075 (L strain); 1511840+1511919 (S strain). The *KANMX* cassette was then replaced by HR using CRISPR/Cas9 (pEF526) with the amplified donor sequence mCherry* containing cutting site of I-SceI (5'-ttacgctagggataacagggtaatatagcg-3'), from pCM189-mCherry*-I-SceI (cs), with oligos oEA310/311. The dCen (pEF573) plasmid was obtained by PCR amplifying mCherry sequence with (oEA307/308) oligonucleotides and cloned into pRS412 (*ADE2* centromeric plasmid) digested with *BamH*I and *EcoR*I. pB07 plasmid containing I-SceI under the control of a *GAL1-10* promoter in a *HIS3* plasmid was obtained from B Pardo.

The H2A-S129A and H2A-S129E mutants were constructed as described in *García Fernández et al., 2021*. Briefly, the CRISPR/Cas9 containing plasmids (pEF567 and pEF568) including the guide RNA targeting *HTA1* (oEA325/326) and *HTA2* (oEA327/328) inserted into the plasmid pJH2971 (CRISPR/Cas9-KANMX-Addgene plasmid#100955) were used to generate a DNA break, afterward repaired with a donor oligonucleotide of 80 nt carrying the desired mutation. All integrations were confirmed by PCR and mutations checked by sequencing. The primers and RNA guides used are indicated in *supplementary file 3*.

Deletions of *RAD9* (YDR217C) and *RAD51* (YER095W) genes in the different strains were carried out by integration of the amplified *KANMX* cassette derived from the Saccharomyces Genome Deletion Project collection.

## Drop assays

For drop assays, overnight cultures were diluted to a starting density of OD600=1.0 and serial 1:5 or 1:10 dilutions were plated on selective medium (SC-Ade-His) containing raffinose or galactose (2%), and incubated at 30°C for 36 hr.

## Microscopy wide-field conditions

Live cell imaging was done using a wide-field microscopy system featuring a Nikon Ti-E body equipped with the Perfect Focus System and a ×60 oil immersion objective with a numerical aperture of 1.4 (Nikon, Plan APO). We used an Andor Neo sCMOS camera, which features a large field of view of 276×233 µm$^2$ at a pixel size of 108 nm. We acquired 3 min films consecutively with an exposure time of 100 ms. The complete imaging system including camera, piezo, LEDs (SpectraX) was controlled by the NIS element software (version 4.60).

## Image analysis and statistics

Particle tracking and MSD analyses were performed as described in *García Fernández et al., 2021*. Briefly, a spot-tracking algorithm (Fiji macro) allows isolating each spot and extracting its trajectory from the 2D time-lapse sequences. MATLAB scripts are then applied to correct global displacements,

control signal-noise ratio, and compute MSDs curves for each trajectory using non-overlapping time intervals and power laws fitting to population-averaged MSD over 10 s intervals qPCR and RT-qPCR.

Cells were grown to exponential phase in raffinose selective medium and induced by addition of galactose 2% final concentration. 1–2·10⁸ cells were centrifuged for 1 min at 13,000 rpm and 300 mg of glass micro beads, 200 µL of lysis solution (2% Triton X-100, 1% SDS, 100 mM NaCl, 10 mM Tris pH 8, 1 mM EDTA) and 200 µL of phenol-chloroform-isoamyl alcohol were added to the cell pellet. The mixture was vortexed for 10 min at 4°C and centrifuged for 10 min at 13,000 rpm; 140 µL of the aqueous phase were transferred into new tubes containing 500 µL of cold 100% ethanol. The mixture was centrifuged for 10 min at 13,000 rpm at 4°C, the pellet washed with 500 µL of 70% ethanol, dried at room temperature, resuspended in 40 µL of TE containing 20 µg/mL of RNAse and incubated at 42°C for 30 min. The DNA was quantified by SimpliNano from Biochrom.

For RT-qPCR, RNA was extracted with the Machery Nagel kit, quantified by Nanoview and absence of RNA degradation checked by gel electrophoresis migration. Maxima First Strand was used for RT reaction on 1.5 µg RNA.

The qPCR was performed on LightCycler 480 (Roche) on a 96-well plate under the following conditions: pre-incubation (95°C – 5 min – 4.4 °C/min), amplification (45 cycles; 95°C – 10 s - 4.4°C/min; 58°C – 10 s - 2.2 °C/min; 72°C – 20 s – 4,4 °C/min), melting curve (95°C – 5 s - 4.4 °C/min; 60°C – 1 min – 2.2 °C/min; 97°C – continuous – 0.06 °C/min; 10 acquisitons/°C) and cooling (40°C – 30 s - 2.2 °C/min.).

The primers oEA312/313 (*supplementary file 3*) were used for qPCR and the actin gene as amplification control.

## Chromatin immunoprecipitation

This protocol is adapted from *Forey et al., 2020*, 1×10⁹ cells were cross-linked for 10 min with 1% formaldehyde (Sigma F8775) at room temperature on a shaking device. Fixation was quenched by addition of 0.25 M glycine (Sigma G8898) for 5 min under agitation. Cells were washed two times with cold TBS1X (4°C). Dry pellets were frozen and stored at –20°C. Cell pellets were resuspended in lysis buffer (50 mM HEPES-KOH pH 7.5, 140 mM NaCl, 1 mM EDTA, 1% Triton-X100, 0.1% sodium deoxycholate) supplemented with 1 mM PMSF, phosphatase inhibitor, and anti-protease (complete Tablet, Roche, 505649001) and lysed three times by beads shaking. The lysate (WCE, Whole Cell Extract) volume was sonicated with a Q500 sonicator (Qsonica; 3 cycles: 40 s ON, 40 s OFF, amplitude medium). Twenty µL of input material were saved for qPCR. Approximately 180 µL (13 OD) of the input were incubated with 0.5% BSA, 1 µL of DNA carrier (10 mg/mL) and 2 µL of anti-γ-H2A(X) (Abcam 15083) on a rotating wheel overnight at 4°C. The day after, 30 µL of protein G Sepharose beads were washed three times and resuspended in 90 µL final of lysis buffer, which were added to the overnight culture during 2 hr on a rotating wheel at 4°C. Beads were then collected and washed on ice with cold solutions: twice with Lysis buffer (50 mM HEPES-KOH pH 7.5, 140 mM NaCl, 1 mM EDTA, 1% Triton X-100, 0.1% sodium deoxycholate), twice with Lysis buffer added with 360 mM NaCl, twice with Washing buffer (10 mM Tris-HCl pH 8, 0.25 M LiCl, 0.5% IGEPAL, 1 mM EDTA, 0.1% sodium deoxycholate) and once with TE (10 mM Tris-HCl pH 8, 1 mM EDTA). Antibodies were uncoupled from beads with 150 µL of Elution buffer (50 mM Tris-HCl pH 8, 10 mM EDTA, 1% SDS) for 20 min at 65°C. Eluates were incubated with 120 µL of TE containing 0.1% SDS at 65°C for 6 hr to de-crosslink. Then 130 µL of TE containing 60 µg RNase A (Sigma, R65-13) were added and the samples were incubated for 2 hr at 37°C. Proteins were digested by addition of 20 µL of proteinase K (Sigma, P6556) at 20 mg/mL to the samples and incubation for 2 hr at 37°C. DNA purification was completed with the QIAquick PCR Purification Kit (n28104). Finally, qPCRs were performed in a LightCycler480 (Roche). IP/input ratios were calculated and qPCR results were normalized on ChIP-qPCR Act1 for γ-H2A(X). For each strain, three independent experiments were performed with the corresponding controls.

## FACS

For cell cycle, cells were grown to mid-log-phase in liquid cultures, and treated or not with Zeocin at 250 mg/µL during 6 hr at 30°C. After incubation, samples were fixed with 70% ethanol and kept at 4°C for 48 hr. Cells were then resuspended in 50 mM sodium citrate (pH 7) containing RnaseA at 0.2 mg/mL final concentration. After incubation at 37°C for 1 hr, Sytox Green was added to a final concentration of 1 mM. A total of 10⁶ cells were analyzed with a CANTO II flow cytometer (BD Biosciences).

Aggregates and dead cells were gated out, and percentages of cells with 1C and 2C DNA content were calculated using FLOWJO software.

For the kinetics of fluorescence red cells, $1–2 \cdot 10^7$ cells were removed and centrifuged. All the following steps were done in the dark. The pellet was washed once with cold PBS at 4°C. After removal of the supernatant, the pellet was resuspended in 500 µL of cold PBS and 500 µL of cold PBS with formaldehyde (to a final concentration of 1%) were added and gently mixed. The cells were incubated for 15 min at 4°C. The cells were then washed three times with sodium citrate (50 mM) and prepared at a concentration of $1.10^6$ cells/mL for the cytometry carried out immediately after (in order to limit the loss of fluorescence). Following steps were performed as above.

## Acknowledgements

We thank Benjamin Pardo for his advices in setting up the ChIP protocol and Stanford and Sup BioTech internships Tara Shanon, Dahee Chung, and Noémie Guerre for their participation in different constructs steps of this study. We acknowledge the critical and constructive reading of Amandine Bonnet, Judith Miné-Hattab, and Pascale Lesage, and thank Adeline Veillet and Gerjan Laenen for their help in proofreading and editing the English version of the manuscript. We thank Jean Michel Arbona and the team members for numerous discussions.This research was funded by the Agence Nationale de la Recherche (ANR-17-CE11-0025 to EF), the initiatives d'excellence (Idex ANR11-LABX-0071, IDEX-0005-), the Institut National du Cancer (INCA), grant number PLBIOR21018HH and Fondation ARC pour la recherche sur le Cancer, grants number PJA32020060002313. FGF acknowledges the Peruvian Scholarship Cienciactiva of Consejo Nacional de Ciencia, Tecnología e Innovación Tecnológica (CONCYTEC) for supporting her PhD study at INSERM and Diderot University and support by Fondation ARC pour la Recherche sur le Cancer (DOC20190508798). EA acknowledges the IDEX USPC (NUPGE15RDX), ACS and YK the Fondation pour la Recherche Médicale FRM (ECO202006011576 and ING20160435205, respectively).

## Additional information

### Funding

| Funder | Grant reference number | Author |
|---|---|---|
| Agence Nationale de la Recherche | ANR-17-CE11-0025 | Emmanuelle Fabre |
| Agence Nationale de la Recherche | Idex ANR11-LABX-0071 | Emmanuelle Fabre |
| Institut National Du Cancer | PLBIOR21018HH | Emmanuelle Fabre |
| Fondation ARC pour la Recherche sur le Cancer | PJA32020060002313 | Emmanuelle Fabre |
| Fondation ARC pour la Recherche sur le Cancer | DOC20190508798 | Fabiola García Fernández |
| Consejo Nacional de Ciencia, Tecnología e Innovación Tecnológica | | Fabiola García Fernández |
| Fondation pour la Recherche Médicale | ECO202006011576 | Ànnia Carré Simon |
| Fondation pour la Recherche Médicale | ING20160435205 | Yasmine Khalil |
| Initiative d'Excellence IDEX USPC | NUPGE15RDX | Etienne Almayrac |
| Agence Nationale de la Recherche | IDEX-0005 | Emmanuelle Fabre |

| Funder | Grant reference number | Author |
|---|---|---|

The funders had no role in study design, data collection and interpretation, or the decision to submit the work for publication.

## Author contributions

Fabiola García Fernández, Conceptualization, Data curation, Formal analysis, Validation, Investigation, Visualization, Methodology, Writing – original draft, Writing – review and editing; Etienne Almayrac, Data curation, Formal analysis, Validation, Visualization, Methodology; Ànnia Carré Simon, Data curation, Formal analysis, Validation, Visualization; Renaud Batrin, Formal analysis, Validation, Visualization; Yasmine Khalil, Software, Formal analysis, Visualization; Michel Boissac, Validation, Investigation; Emmanuelle Fabre, Conceptualization, Data curation, Formal analysis, Supervision, Funding acquisition, Validation, Investigation, Visualization, Methodology, Writing – original draft, Project administration, Writing – review and editing

## Author ORCIDs

Fabiola García Fernández http://orcid.org/0000-0003-2909-236X
Renaud Batrin http://orcid.org/0000-0002-5473-315X
Emmanuelle Fabre http://orcid.org/0000-0002-0009-4604

## Ethics

This work does not involve human participants or animal experimentation.

## Decision letter and Author response

Decision letter https://doi.org/10.7554/eLife.78015.sa1
Author response https://doi.org/10.7554/eLife.78015.sa2

# Additional files

## Supplementary files

• Supplementary file 1. Compilation of values extracted from mean squared displacements (MSDs) of the analyzed strains.

• Supplementary file 2. Genotypes of the strains used in this study.

• Supplementary file 3. Oligonucleotides sequences.

• Transparent reporting form

## Data availability

All data generated or analysed during this study are included in the manuscript and Source Data files. The dynamic trajectories analyzed by matlab, have been uploaded to Dryad as well as the scripts to analyze them.

The following dataset was generated:

| Author(s) | Year | Dataset title | Dataset URL | Database and Identifier |
|---|---|---|---|---|
| Fabre E | 2022 | Matlab Files from MSD analyses | https://dx.doi.org/10.5061/dryad.931zcrjn1 | Dryad Digital Repository, 10.5061/dryad.931zcrjn1 |

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
