## [Editor Report]

This study is of relevance to the field of DNA repair. It uses a cleverly designed new recombination assay in yeast to address the impact of DNA break position on global genome mobility. A centromere–proximal DNA double–strand break (DSB) induces an H2A(X) phosphorylation–dependent global mobility that accelerates but is not essential for DSB repair, while a centromere–distal DSB triggers global mobility that is essential for repair and which depends on H2A(X) phosphorylation, Rad9 and Rad51. Together, these data support a model where global genome mobility promotes homologous recombination repair, particularly for centromere–distal DSBs, and help settle some recent controversy in the field.

---

## [Decision Letter]

**Decision letter after peer review:**

Thank you for submitting your article "Global chromatin mobility induced by a double–strand break is dictated by chromosomal conformation and defines the outcome of homologous recombination" for consideration by *eLife*. Your article has been reviewed by 3 peer reviewers, and the evaluation has been overseen by a Reviewing Editor and Kevin Struhl as the Senior Editor. The following individual involved in the review of your submission has agreed to reveal their identity: Aurèle Piazza (Reviewer #3).

Essential revisions:

The reviewers agree that experiments are well–conducted and support the claims and that the manuscript is suitable for publication in *eLife*. They also raise some important points, most of which can be easily addressed by text changes, although some additional controls seem to be required. We highlight what we consider essential revisions here:

1. A common concern is the lack of ability to distinguish between damaged (80%) and undamaged (20%) cells for MSD experiments. Addressing this point requires additional experiments, and a possible approach to address this point is proposed by reviewer #3 (although other approaches might be used).

2. The discussion should directly focus on the proteins tested (see comments from reviewer #1). It should also highlight the limitations of the system used (caveats of a plasmid–based system).

3. Presentation of statistics, sample sizes, etc should be easy to identify within each figure (see comments from reviewer #1).

4. Reviewer #2 raises an important point that even after most of the break is likely repaired, there is still elevated mobility. While establishing mobility at even later time points might be outside the scope of this manuscript, this is a point worth discussing.

5. In the absence of Rad51, mobility still occurs in the Cen–proximal cut strain. It is important to show HR efficiency in this strain, as pointed out by reviewer #2.

6. Reviewer #3 points out interesting observations ("painting model") that the authors might find helpful to integrate in the model/discussion. There are also some issues pointed out by this reviewer about consistency across experiments, which we think should be mostly addressable with text changes, and by acknowledging the different sensitivities of distinct assays. Additionally, how the HR defects of checkpoint mutants might affect

7. A discussion of the data and model fit with previous work (see point 3 by reviewer #3, and the reference to studies from the Rothstein and Gasser labs) should be provided.

*Reviewer #1 (Recommendations for the authors):*

General feedback:

– Although this study focuses on global mobility effects, it is reasonable to think that the global mobility effects reported here occur in parallel to DSB site–targeted mobility. The introduction and especially the discussion would benefit from explaining how the authors envision the global mobility processes reported here intersecting with the DSB–targeted mobility that is well documented in the literature.

– In the model and discussion, highlighting Mec1 as opposed to H2A, Rad9, Rad51 is puzzling. An area for improvement for the manuscript would be to improve the discussion and model to focus on factors directly tested in the study.

– Another area for improvement centers around the presentation of error bars, sample sizes, and statistics in a manner that does not necessitate that the reader cross–reference the figure, main text, figure legend, and supplementary tables to determine exactly what was done in a given data panel. Especially complete and clarify the information on sample sizes, replicates, and statistics in the figure legends.

– The discussion should include a section on the limitations of the systems used and especially the caveats associated with the use of plasmid–based donor sequences in most experiments.

Presentation and clarity:

– Figure 2B and other MSD plots: error bars needed to interpret the MSD data accurately are absent; also, the lines in some of the MSD plots are so faded that this reviewer struggled to distinguish some of them. Please consider making the lines thicker or changing their colors.

– Figure 4A and 4B: p values are only shown for one of the four graphs when it seems that large and likely statistically significant changes were observed in the other three graphs. Please clarify.

– Callout to Figure 4E, S, grey bars near the bottom of page 9 of the text: should this refer to Figure 4D instead of 4E?

– Page 12, 2nd paragraph, 2nd line: "end of the IVR chromosome," should be "VR"?

– Page 3, paragraph 2: "that allows simultaneously tracking".

– Overall, the paper would benefit from detailed English language editing.

*Reviewer #2 (Recommendations for the authors):*

It would be valuable to show in the supplemental Figure 1 the frequency of red cells at all of the timepoints with the empty vector (no repair template) both with and without DSB induction to show the false positive rate for red forming cells. In other words, what is the background level of red–forming cells that result from the reversion of the stop codon both before and after DSB induction?

Please discuss why the experiments shown in Figure 1 were not performed in the strain harboring the tetO repeats that were used for visualization and MSD analysis. The authors did a great job of classifying repair dynamics using their THRIV system, but then add another component to it (V–Vis) before performing the crucial MSD experiments. Is it because the GFP signal would interfere with the FACS analysis? Maybe the authors can at least do the spot assay as quality control?

It is not clear why 6 hours was chosen as the time to use for DSB induction. Is it due to the delay observed between the I–SceI enzyme expression and an effective cutting? If so, please state the rationale. Do 3 hours of DSB induction change the results observed in Figure 2?

The results shown in Figure 3 are a highlight of the paper. The authors do a great job utilizing the strength of their system to examine how mobility changes at different stages of the repair process (beginning vs end). However, it is surprising that even after 12 hours of DSB induction and clear evidence of completed repair (red cell), there is still elevated levels of mobility when compared to –DSB control. If repair is complete, why is there still elevated mobility? What are the results of mobility at 24 hours after DSB induction (the last timepoint taken in Figure 1)? How long does it take for the mobility response to completely return to baseline after HR is successful? Please discuss these points.

For Figure 5, were the HR kinetics examined in the rad51 mutant using the FACS assay? Does HR repair occur normally in the C strain when rad51 is deleted given that the global mobility response is shown to still occur in that strain?

*Reviewer #3 (Recommendations for the authors):*

– Experiments in Figure 2C rule out that increased mobility of undamaged sites is due to a centromere detachment from the SPB, and instead suggests a spreading mechanism. Authors should spell it out clearly in the text.

– Lay out more clearly the "painting" model.

– Figure 3A–B: red cells have repaired, and are not in the same cell cycle phase (see figure 1D FACS and picture of the red cells), which is known to affect mobility (Cheblah Gasser 2020).

– MSDs without DSB are different in Figure 2B and in Figure 2C (0.05 vs 0.03 at 10 seconds). Why? The same thing with C–DSB and C+DSB, values are different in Figure 2B and 2C.

– Indicate at which time post–DSB induction the MSD is monitored on the figure panel.

– Inconsistent spot assays across figures for the same condition, or similar conditions (Figure 1E, Figure S3B, Figure S4B). For instance, C strain WT somewhat grows on +Gal in Figure 1E and S4B, but not in S3B. The S+dCen strain WT +DSB is dead in Figure 1E but shows excellent recovery in S3B and S4B. There are other inconsistencies within Figure S4B, with C and S strains without DSB showing different growth despite having the same genotype. These spot assays should be redone in parallel, with the same dilution and let grow the same amount of time. The methods indicate that the plating is made from a saturating culture. It is standard practice to plate exponentially growing cells, which may also improve data reproducibility.

– Figure S2c: colors are switched in one panel.

[Editors' note: further revisions were suggested prior to acceptance, as described below.]

Thank you for resubmitting your work entitled "Global chromatin mobility induced by a double–strand break is dictated by chromosomal conformation and defines the outcome of homologous recombination" for further consideration by *eLife*. Your revised article has been evaluated by Kevin Struhl (Senior Editor) and a Reviewing Editor, in consultation with the reviewers.

The manuscript has been significantly improved but there are some remaining issues that need to be addressed, as outlined below:

1. The addition of text comparing the genetic requirements of local vs. global mobility has improved the paper. Yet, how the authors envision DSB site–focused molecular mechanisms intersecting, or not, with the local vs. global phenomena reported here remains somewhat unclear. Such molecular mechanisms in yeast include the release of centromeres from the SPB (global), repair protein phase separation (local), microtubule/motor–dependent DSB mobility (local), and DSB/global chromatin changes or condensation (local/global). How may such molecular mechanisms intersect with the proximal vs. distal homology systems used in this study? Adding a paragraph to discuss the potential intersection of such mechanisms, or lack thereof, to the discussion would further enrich the discussion/paper.

2. Limitations related to the common failure of MSD to detect directed mobility and to over–represent random mobility should be added to the limitations section. How this limitation may affect the interpretation of the MSD curves and conclusions in this study should be mentioned.

3. We encourage the authors to include the scatter plot showing the MSD for the 'damaged–only', the 'repaired–only', or the 20%–80% mixed population in the supplementary figures, and include a brief discussion about it in the text.

---

## [Author Response]

Essential revisions:The reviewers agree that experiments are well–conducted and support the claims and that the manuscript is suitable for publication in eLife. They also raise some important points, most of which can be easily addressed by text changes, although some additional controls seem to be required. We highlight what we consider essential revisions here:1. A common concern is the lack of ability to distinguish between damaged (80%) and undamaged (20%) cells for MSD experiments. Addressing this point requires additional experiments, and a possible approach to address this point is proposed by reviewer #3 (although other approaches might be used).

The referees raise the important question of the heterogeneity of a population, a central challenge in biology. Here we wish to distinguish between undamaged and damaged cells. Even if a selection of the damaged cells had been made, this would not solve entirely the inherent cell to cell variation: at a given time, it is possible that a cell, although damaged, moves little and conversely that a cell moves more, even if not damaged. The question of heterogeneity is therefore important and the subject of intense research that goes beyond the framework of our work (Altschuler and Wu, 2010). However, in order to start to clarify if a bias could exist when considering a mixed population (20% undamaged and 80% damaged), we analyzed MSDs, using a scatter plot. We considered two population of cells where the damage is the best controlled, i.e. (i) the red population which we know has been repaired and, importantly, has lost the cut site and will be not cut again (undamaged - only population) and (ii) the white population, blocked in G2/M, because it is damaged and not repaired (damaged - only population). These two populations show very significant differences in their median MSDs. We artificially mixed the MSDs values obtained from these two populations at a rate of 20% of undamaged - only cells and 80% of damaged - only cells. We observed that the mean MSDs of the damaged - only and undamaged - only cells were significantly different. Yet, the mean MSD of damaged - only cells was not statistically different from the mean MSD from the 20% - 80% mixed cell population. Thus, the conclusions based on the average MSDs of all cells remain consistent.

2. The discussion should directly focus on the proteins tested (see comments from reviewer #1). It should also highlight the limitations of the system used (caveats of a plasmid–based system).

We have rewritten parts of the discussion to better reflect the proteins tested. In particular, our experiments allow drawing a parallel between the local mobility and the distal mobility because both require the Rad51 filament and the control of Rad9.

We have added a paragraph called Systems Limitations which include this particular aspect; although the THRIV system was validated also with a chromosomal donor, the plasmid/chromosome contexts are different, and may in particular impact the time or residency of the nucleoprotein filament, as suggested by Reviewer 3.

3. Presentation of statistics, sample sizes, etc should be easy to identify within each figure (see comments from reviewer #1).

We have now added these data in the legend of each figure.

4. Reviewer #2 raises an important point that even after most of the break is likely repaired, there is still elevated mobility. While establishing mobility at even later time points might be outside the scope of this manuscript, this is a point worth discussing.

It is noteworthy that at 12h, red cells that have lost the cleavage site for I - SceI and thus can no longer be cut by the enzyme, still show a higher mobility than the undamaged control. The most probable hypothesis is that the H2A phosphorylation has not yet returned to normal at this time point. There are several mechanisms to loose H2A phosphorylation that include exchange of the phosphorylated histone and its possible degradation or dephosphorylation of H2A. Kinetics of H2A dephosphorylation by ChIP and/or measurement of chromatin dynamics in backgrounds mutated for H2A dephosphorylation, should be performed to confirm this hypothesis, experiments that are beyond the scope of this manuscript. However, it is known that while H2A dephosphorylation around the DSB is quite fast (~1h), H2A dephosphorylation at 20kb from the DSB requires more time (~5h) and foci of γH2A are still detectable at ~6h after damage (Keogh et al., 2006). On the other hand, kinetics of dephosphorylation in trans are not known. Interestingly, this study demonstrated that H2A dephosphorylation might not be complete even at the end of repair, which is consistent with the maintenance of γH2A on chromatin, including in red cells. These red cells could therefore, still exhibit increased mobility.

5. In the absence of Rad51, mobility still occurs in the Cen–proximal cut strain. It is important to show HR efficiency in this strain, as pointed out by reviewer #2.

In the C strain, when the DSB is induced close to the centromere, cells deleted for Rad51 do not grow, even in the presence of the donor, although an increase in mobility is detected. Poor growth is a sign of ineffective repair that impedes FACS analyses for calculation of HR efficiency. We propose that mobility promotes recombination, but cannot supplant such essential recombination proteins as Rad51 in the recombination reaction.

6. Reviewer #3 points out interesting observations ("painting model") that the authors might find helpful to integrate in the model/discussion. There are also some issues pointed out by this reviewer about consistency across experiments, which we think should be mostly addressable with text changes, and by acknowledging the different sensitivities of distinct assays. Additionally, how the HR defects of checkpoint mutants might affect

The paintbrush model was very useful in explaining the distal mobility that indeed is linked to local mobility genetic requirements. It is also helpful to think of a different model than the paintbrush model when pericentromeric damage occurs. To stay in the terms of painting technique, this latter model would be similar to the pouring technique, when oil paint is deposited on water and spreads in a multidirectional manner. It is likely that Mec1 or Tel1 are the factors responsible for this spreading pattern. We therefore propose to maintain the notion of two distinct types of mobilities. Without going into pictorial techniques in the text, we have attempted to clarify these two models in the manuscript.

We have corrected the experimental inconsistencies noted by referee 3, mainly due to different incubation times.

7. A discussion of the data and model fit with previous work (see point 3 by reviewer #3, and the reference to studies from the Rothstein and Gasser labs) should be provided.

We have added a new paragraph in the discussion to better account for references to previous studies by Rothstein and Gasser labs.

Reviewer #1 (Recommendations for the authors):General feedback:– Although this study focuses on global mobility effects, it is reasonable to think that the global mobility effects reported here occur in parallel to DSB site–targeted mobility. The introduction and especially the discussion would benefit from explaining how the authors envision the global mobility processes reported here intersecting with the DSB–targeted mobility that is well documented in the literature.

We thank the reviewer for this constructive criticism, because indeed, the parameters that govern the distal mobility that we document are the same as those of the local mobility, highlighting the parallel mechanisms for both mobilities and therefore deserve to be better explained and discussed. This is now done.

– In the model and discussion, highlighting Mec1 as opposed to H2A, Rad9, Rad51 is puzzling. An area for improvement for the manuscript would be to improve the discussion and model to focus on factors directly tested in the study.

The mechanism of mobility around the centromere, related to the propagation of H2A phosphorylation is intriguing and could be explained by the mode of diffusion of Mec1 or Tel1. As pointed out by the reviewer, the involvement of the Rad51 nucleoprotein filament and of Rad9 in the global mobility induced when the damage is far from the centromere, according to a mechanism similar to that of the local mobility, deserves to be better highlighted with a more precise discussion of these two proteins. We have therefore added this aspect to our discussion.

– Another area for improvement centers around the presentation of error bars, sample sizes, and statistics in a manner that does not necessitate that the reader cross–reference the figure, main text, figure legend, and supplementary tables to determine exactly what was done in a given data panel. Especially complete and clarify the information on sample sizes, replicates, and statistics in the figure legends.

We have completed as much as possible the legends with the data of sample sizes, replicates and statistics in order to simplify the reading.

– The discussion should include a section on the limitations of the systems used and especially the caveats associated with the use of plasmid–based donor sequences in most experiments.

We have added a paragraph called Limitations of the systems used.

As stated in the general comment, although the THRIV system was validated also with a chromosomal donor, the plasmid/chromosome contexts are different, and may in particular impact the time or residency of the nucleoprotein filament, as suggested by Reviewer 3.

Presentation and clarity:– Figure 2B and other MSD plots: error bars needed to interpret the MSD data accurately are absent; also, the lines in some of the MSD plots are so faded that this reviewer struggled to distinguish some of them. Please consider making the lines thicker or changing their colors.

As the reviewer noticed, the MSDs curves were not very distinct and the standard deviations were not really visible; we have made the lines thicker and the sd’s more visible

– Figure 4A and 4B: p values are only shown for one of the four graphs when it seems that large and likely statistically significant changes were observed in the other three graphs. Please clarify.

Pvalues are now added when possible.

– Callout to Figure 4E, S, grey bars near the bottom of page 9 of the text: should this refer to Figure 4D instead of 4E?

Indeed, the error is now corrected.

– Page 12, 2^nd^ paragraph, 2^nd^ line: “end of the IVR chromosome,” should be “VR”?

It is the labelled and tracked locus which is located on the right arm of chromosome V. The I – SceI cutting sites discussed in this paragraph are located on the right arm of chromosome IV.

– Page 3, paragraph 2: “that allows simultaneously tracking”.

Replaced by « that allows simultaneously to monitor…”.

– Overall, the paper would benefit from detailed English language editing.

We had the text proofread and corrected by a native English speaker.

Reviewer #2 (Recommendations for the authors):It would be valuable to show in the supplemental Figure 1 the frequency of red cells at all of the timepoints with the empty vector (no repair template) both with and without DSB induction to show the false positive rate for red forming cells. In other words, what is the background level of red–forming cells that result from the reversion of the stop codon both before and after DSB induction?

If we have not indicated in Figure 1F the % of red colonies in the absence of donor it is because it remains constant and low over time around 0.55% ± 0.004 and 1.02% ± 0.85 for the C and S strains, respectively. We have added in Figure Supp.1C quantifications and examples of cells after 24h in the C and S strains in the absence of donor to illustrate our point.

Please discuss why the experiments shown in Figure 1 were not performed in the strain harboring the tetO repeats that were used for visualization and MSD analysis. The authors did a great job of classifying repair dynamics using their THRIV system, but then add another component to it (V–Vis) before performing the crucial MSD experiments. Is it because the GFP signal would interfere with the FACS analysis? Maybe the authors can at least do the spot assay as quality control?

We apologize if our description of the strains was not sufficiently explicit. These are the same strains in Figure 1 and Figure 2. We had split the description into 2 parts for clarity, but we now state from Figure 1 that these strains contain the cutting sites for I - SceI, the donor sequence and the visualization system.

It is not clear why 6 hours was chosen as the time to use for DSB induction. Is it due to the delay observed between the I–SceI enzyme expression and an effective cutting? If so, please state the rationale. Do 3 hours of DSB induction change the results observed in Figure 2?

We chose 6h, because it is the induction time for which we have the best compromise, on the number of damaged cells and the beginning of the appearance of red cells. At 3h, as the referee pointed out, the enzyme is expressed but the cutting is not yet effective. The rationale of the long - time is now discussed in the “limitations of the system” paragraph.

The results shown in Figure 3 are a highlight of the paper. The authors do a great job utilizing the strength of their system to examine how mobility changes at different stages of the repair process (beginning vs end). However, it is surprising that even after 12 hours of DSB induction and clear evidence of completed repair (red cell), there is still elevated levels of mobility when compared to –DSB control. If repair is complete, why is there still elevated mobility? What are the results of mobility at 24 hours after DSB induction (the last timepoint taken in Figure 1)? How long does it take for the mobility response to completely return to baseline after HR is successful? Please discuss these points.

We thank the referee for the appreciation of this difficult experiment, which was indeed very instructive. Concerning the dynamics of the red cells at 12h, the most probable hypothesis is that the H2A phosphorylation has not yet returned to normal at this time point. There are several mechanisms to loose H2A phosphorylation that include exchange of the phosphorylated histone and its possible degradation or dephosphorylation of H2A. Kinetics of H2A dephosphorylation by ChIP and/or measurement of chromatin dynamics in backgrounds mutated for H2A dephosphorylation, should be performed to confirm this hypothesis, experiments that are beyond the scope of this manuscript. However, it is known that while H2A dephosphorylation around the DSB is quite fast (~1h), H2A dephosphorylation at 20kb from the DSB requires more time (~5h) and foci of γH2A are still detectable at ~6h after damage (Keogh et al., 2006). On the other hand, kinetics of dephosphorylation in trans are not known Interestingly, this study demonstrated that H2A dephosphorylation may not be complete even at the end of repair, which is consistent with the maintenance of γH2A on chromatin, including in red cells. These red cells could therefore, still exhibit increased mobility.

For Figure 5, were the HR kinetics examined in the rad51 mutant using the FACS assay? Does HR repair occur normally in the C strain when rad51 is deleted given that the global mobility response is shown to still occur in that strain?

As stated in the general comment, in the C strain, when the DSB is induced close to the centromere, cells deleted for Rad51 grow poorly, even in the presence of the donor, although an increase in mobility is detected. Poor growth is a sign of ineffective repair that impedes FACS analyses for calculation of HR efficiency. We propose that mobility promotes recombination, but cannot supplant such essential recombination proteins as Rad51 in the recombination reaction.

Reviewer #3 (Recommendations for the authors):– Experiments in Figure 2C rule out that increased mobility of undamaged sites is due to a centromere detachment from the SPB, and instead suggests a spreading mechanism. Authors should spell it out clearly in the text.

In our opinion, it is still difficult at this stage to estimate whether or not pericentromeric DSB causes microtubule detachment from kinetochores. It remains possible that the spreading of γH2A weakens this attachment, but only super - resolution microscopy experiments could perhaps resolve this point.

– Lay out more clearly the "painting" model.

The model related to the formation of the Rad51 filament and the probable residence time of this filament with the donor as well as the propagation pattern of γH2A are now better highlighted, in the text and mentioned in the legend of figure 6.

– Figure 3A–B: red cells have repaired, and are not in the same cell cycle phase (see figure 1D FACS and picture of the red cells), which is known to affect mobility (Cheblah Gasser 2020).

The distinction between white cells, blocked in G2/M, and cycling red cells effectively highlights the fact that the increased dynamics may be related to a cycling defect. However, it is possible to observe an increase in dynamics in a Rad9 mutant, where the cycle is no longer blocked upon damage (Figure 3C and our study Garcia Fernandez, 2021), suggesting that G2/M block is not sufficient to explain an increase in dynamics.

– MSDs without DSB are different in Figure 2B and in Figure 2C (0.05 vs 0.03 at 10 seconds). Why? The same thing with C–DSB and C+DSB, values are different in Figure 2B and 2C.

The differences between MSDs experiments are a good example of biological heterogeneity. This is the reason why in each experiment the pictures are rigorously taken in the control strains without DSB and compared to the strains having undergone the DSB.

– Indicate at which time post–DSB induction the MSD is monitored on the figure panel.

In the legend, 6h.

– Inconsistent spot assays across figures for the same condition, or similar conditions (Figure 1E, Figure S3B, Figure S4B). For instance, C strain WT somewhat grows on +Gal in Figure 1E and S4B, but not in S3B. The S+dCen strain WT +DSB is dead in Figure 1E but shows excellent recovery in S3B and S4B. There are other inconsistencies within Figure S4B, with C and S strains without DSB showing different growth despite having the same genotype. These spot assays should be redone in parallel, with the same dilution and let grow the same amount of time. The methods indicate that the plating is made from a saturating culture. It is standard practice to plate exponentially growing cells, which may also improve data reproducibility.

The spot assay was done on cells in exponential phase, we have corrected this error in the methods. We thank the referee for his critical eye, the differences were due to the growth times and the dilution between the different figures (1:10 dilutions 48h for Figure 1, 1:5 60h for S4B, 40h for S3B) and to the different contrast of that picture. The reason for a longer time is related to the slow growth of the SA mutant. Times and dilutions are now indicated in the legend. All strains grow similarly between experiments.

– Figure S2c: colors are switched in one panel.

Corrected

References

Altschuler SJ, Wu LF. 2010. Cellular Heterogeneity: Do Differences Make a Difference? Cell 141:559–563. doi:10.1016/j.cell.2010.04.033

Cheblal A, Challa K, Seeber A, Shimada K, Yoshida H, Ferreira HC, Amitai A, Gasser SM. 2020. DNA Damage - Induced Nucleosome Depletion Enhances Homology Search Independently of Local Break Movement. Mol Cell 80:311 - 326.e4. doi:10.1016/j.molcel.2020.09.002

Garcia Fernandez F, Lemos B, Khalil Y, Batrin R, Haber JE, Fabre E. 2021. Modified chromosome structure caused by phosphomimetic H2A modulates the DNA damage response by increasing chromatin mobility. J Cell Sci jcs.258500. doi:10.1242/jcs.258500

Hauer MH, Seeber A, Singh V, Thierry R, Sack R, Amitai A, Kryzhanovska M, Eglinger J, Holcman D, Owen - Hughes T, Gasser SM. 2017. Histone degradation in response to DNA damage enhances chromatin dynamics and recombination rates. Nat Struct Mol Biol. doi:10.1038/nsmb.3347

Herbert S, Brion A, Arbona J - M, Lelek M, Veillet A, Lelandais B, Parmar J, Fernández FG, Almayrac E, Khalil Y, Birgy E, Fabre E, Zimmer C. 2017. Chromatin stiffening underlies enhanced locus mobility after DNA damage in budding yeast. EMBO J 36. doi:10.15252/embj.201695842

Keogh MC, Kim JA, Downey M, Fillingham J, Chowdhury D, Harrison JC, Onishi M, Datta N, Galicia S, Emili A, Lieberman J, Shen X, Buratowski S, Haber JE, Durocher D, Greenblatt JF, Krogan NJ. 2006. A phosphatase complex that dephosphorylates γH2AX regulates DNA damage checkpoint recovery. Nature 439:497–501. doi:10.1038/nature04384

Smith MJ, Bryant EE, Rothstein R. 2018. Increased chromosomal mobility after DNA damage is controlled by interactions between the recombination machinery and the checkpoint. Genes Dev 32:1242–1251. doi:10.1101/gad.317966.118

[Editors' note: further revisions were suggested prior to acceptance, as described below.]

The manuscript has been significantly improved but there are some remaining issues that need to be addressed, as outlined below:1. The addition of text comparing the genetic requirements of local vs. global mobility has improved the paper. Yet, how the authors envision DSB site–focused molecular mechanisms intersecting, or not, with the local vs. global phenomena reported here remains somewhat unclear. Such molecular mechanisms in yeast include the release of centromeres from the SPB (global), repair protein phase separation (local), microtubule/motor–dependent DSB mobility (local), and DSB/global chromatin changes or condensation (local/global). How may such molecular mechanisms intersect with the proximal vs. distal homology systems used in this study? Adding a paragraph to discuss the potential intersection of such mechanisms, or lack thereof, to the discussion would further enrich the discussion/paper.

Our work highlights two types of global (proximal/distal) mobilities. As proposed, we have incremented the discussion by a new paragraph.

Indeed, discussion of the mechanistic similarities between local and global (distal) mobilities highlights the strength of existing mechanisms to initiate mobility and facilitate repair. Moreover, the possible involvement of cytoskeletal factors in these mobilities broadens the discussion and opens an interesting field of future research.

2. Limitations related to the common failure of MSD to detect directed mobility and to over–represent random mobility should be added to the limitations section. How this limitation may affect the interpretation of the MSD curves and conclusions in this study should be mentioned.

We have added this limitation in the corresponding section.

3. We encourage the authors to include the scatter plot showing the MSD for the 'damaged–only', the 'repaired–only', or the 20%–80% mixed population in the supplementary figures, and include a brief discussion about it in the text.

We have added this text p7 lines 192 - 202, included the corresponding figure in Supp Figure 3A along with its legend.